fluid mechanics/applied mathematics/biomechanics

ideal fluids, slender body theory, anguilliform swimming, hydrodynamic forces, sea snake

**Author for correspondence:**
Gil Iosilevskii
e-mail: igil@technion.ac.il

# Hydrodynamics of a twisting slender swimmer

Gil Iosilevskii and Alexander Rashkovsky

Faculty of Aerospace Engineering, Technion, Haifa 32000, Israel

GI, 0000-0002-4114-3214

Sea snakes propel themselves by lateral deformation waves moving backwards along their bodies faster than they swim. In contrast to typical anguilliform swimmers, however, their swimming is characterized by exaggerated torsional waves that lead the lateral ones. The effect of torsional waves on hydrodynamic forces generated by an anguilliform swimmer is the subject matter of this study. The forces, and the power needed to sustain them, are found analytically using the framework of the slender (elongated) body theory. It is shown that combinations of torsional waves and angle of attack can generate both thrust and lift, whereas combinations of torsional and lateral waves can generate lift of the same magnitude as thrust. Generation of lift comes at a price of increasing tail amplitude, but otherwise carries practically no energetic penalty.

## 1. Introduction

Sea snakes have flattened bodies with no fins, and they propel themselves by lateral deformation waves moving backwards along their bodies faster than they swim—as a typical eel-like (anguilliform) swimmer does [1]. In contrast to a typical anguilliform swimmer, however, their swimming is characterized by exaggerated torsional waves (their amplitude can exceed 90°) that lead the lateral waves. Can it be that the torsional waves come to balance the swimming snake against gravity? To answer this question, one will need an estimate of hydrodynamic forces acting on an anguilliform swimmer propelling itself by a combination of lateral and torsional waves—these forces are the subject matter of this study.

Hydrodynamic forces acting on an elongated deforming body, moving in a fluid at Reynolds numbers in excess of a few tens of thousands, can be found in several ways. The two extreme approaches are represented by direct numerical solutions of the Navier–Stokes equations [2], and by asymptotic solutions based on the width-to-length ratio of the body as a small parameter and an ideal fluid approximation [3–7]. The last approach, widely known as the elongated (or slender) body theory, allows, at least in principle, to obtain the hydrodynamic forces acting on the body analytically. Preferring simplicity to accuracy, but

deeming the accuracy of the asymptotic approach adequate [8], it is adopted for this study as well. The coherence of the present results within the ideal fluid approximation is furnished in the electronic supplementary material by comparison with numerical simulations based on the vortex lattice method. An indication of their viability is furnished in Appendix I by comparison with observations of a swimming yellow-bellied sea snake *Hydrophis platurus* [1].

The paper is organized in seven sections and 10 short appendices, which contain the details of the underlying derivations. Units, notation, reference frames and the model swimmer are introduced in the next section (§2), and it is where distributed forces acting on the swimmer are derived. Integral forces acting on the swimmer are derived in §3, and further developed in §4 under the assumption that the deformation waves are harmonic. Effects of torsion are analysed in §5. Adequacy of the hydrodynamic forces to balance a swimming snake is assessed in §6. Section 7 concludes the paper.

## 2. Fundamentals

### 2.1. Reference frames

The paper addresses a deformable swimmer of length $l$ that moves, on average, with constant speed $v$ along a straight path in an infinite domain occupied by quiescent fluid of density $\rho$. Throughout the paper, $l$, $v$, $l/v$, $lv$, $\rho v^2$, $\rho v^2 l$, $\rho v^2 l^2$, $\rho v^2 l^3$, $\rho v^3 l$ and $\rho v^3 l^2$ will serve as units of length, velocity, time, potential, pressure, force per unit length, force (or moment per unit length), moment, power per unit length and power, respectively (table 1).

Two adjunct right-handed rectilinear reference frames, C and C′ will be used interchangeably. Both have their $x$-axes opposing the (average) swimming direction, and both follow the swimmer along its average path. C′ is a global (inertial) frame. Its $y$-axis lies in the sagittal plane of the undeformed body and, for the sake of definiteness, points towards its dorsal side. The complementary $z$-axis points left, perpendicular to the sagittal plane. Coordinates of a point relative to C′ will be marked by a prime. Any scalar or vector field parametrized using coordinates of C′ will be marked by a prime as well.

C is a local (non-inertial) reference frame, affixed to each cross section along the body. Its origin is located in the $y'$–$z'$ plane of C′; its $x$-axis passes through the middle of a particular section; and the frame itself is rotated (twisted) about the $x$-axis through angle $\theta(t, x)$, so as to make the $y$-axis pass through the ventral and dorsal edges of that section. Coordinates of a point relative C will remain unmarked, and so will any scalar or vector field parametrized using coordinates of C. Formally,

$$\mathbf{e}_{x'} = \mathbf{e}_x, \tag{2.1}$$

$$\mathbf{e}_{y'} = \mathbf{e}_y \cos \theta(t, x) - \mathbf{e}_z \sin \theta(t, x) \tag{2.2}$$

and
$$\mathbf{e}_{z'} = \mathbf{e}_y \sin \theta(t, x) + \mathbf{e}_z \cos \theta(t, x) \tag{2.3}$$

relate the respective unit vectors;

$$x' = x, \tag{2.4}$$

$$y' = y_0(t, x) - z \sin \theta(t, x) + y \cos \theta(t, x) \tag{2.5}$$

and
$$z' = z_0(t, x) + z \cos \theta(t, x) + y \sin \theta(t, x) \tag{2.6}$$

relate the coordinates. Because of its equivalence with $x$ (equation (2.4)), $x'$ and $x$ will be used interchangeably. By interpretation, $y' = y_0(t, x)$ and $z' = z_0(t, x)$ are equations of the swimmer's centreline in C′.

### 2.2. The model swimmer

An undeformed swimmer is assumed to be flat and of zero thickness.[1] The outline of the swimmer starts with a point at the cranial end ($x = x_n = 0$), and reaches maximal span of $2s_t$ at the caudal end ($x = x_t = 1$).[2]

---

[1]While the shape of its cross section has little or no effect on hydrodynamic forces generated by a non-twisting swimmer [5], it is hardly true for a twisting one. The assumed flatness of the swimmer should be accepted as a limitation of the present model, and extension of its results to a swimmer with a different cross section should be made with due caution.

[2]Ending the model swimmer at the widest section saves the need to address hydrodynamic interaction with its own wake [5,7,8] in an already complex analysis.

**Table 1.** Nomenclature.

| | |
|---|---|
| **fundamental (dimensional) quantities** | |
| $g$ | acceleration of gravity |
| $m$ | body mass |
| $l$ | body length |
| $v$ | swim speed |
| $\rho$ | water density |
| $\nu$ | kinematic viscosity |
| **fundamental units** | |
| $l$ | length |
| $v$ | velocity |
| $l/v$ | time |
| $lv$ | velocity potential |
| $\rho v^2$ | pressure |
| $\rho v^2 l$ | force per unit length |
| $\rho v^2 l^2$ | force, or moment per unit length |
| $\rho v^2 l^3$ | moment |
| $\rho v^3 l$ | power per unit length |
| $\rho v^3 l^2$ | power |
| **non-dimensional quantities** | |
| $A_\pm$ | coefficient with the square-root singularity of $\mu$ at the dorsal and ventral edges; equation (2.37) |
| $B$ | buoyancy; equation (6.4) |
| $C$ | local (non-inertial) reference frame attached to the body of the swimmer |
| $C'$ | global inertial reference frame moving along with the swimmer |
| $C_n$ | standard integral; equation (2.25) |
| $\bar{D}$ | drag coefficient (based on $2\pi s_t^2$ as the reference area); equation (6.1) |
| $\mathbf{e}_x, \mathbf{e}_y, \mathbf{e}_z$ | basis vectors of $C$; positive directions are posterior, dorsal and sinister |
| $\mathbf{e}_{x'}, \mathbf{e}_{y'}, \mathbf{e}_{z'}$ | basis vectors of $C'$; equations (2.1)–(2.3) |
| Fr | Froude number, $\mathrm{Fr} = v/\sqrt{gl}$ |
| $\mathbf{f}$ | force per unit length acting on the swimmer; equation (3.1) |
| $f_{x'}, f_{y'}, f_{z'}$ | components of $\mathbf{f}$ in $C'$; equations (3.3)–(3.5) |
| $f_\pm$ | leading-edge suction (per unit length), dorsal (+) and ventral (−); equation (2.36) |
| $J_n$ | $n$th-order Bessel function of the first kind |
| $k$ | prismatic coefficient; the ratio between the volume of the body and the minimal cylinder enclosing it; equation (6.4) |
| $L$ | lift: the component of the (period averaged) hydrodynamic force along $\mathbf{e}_{y'}$; equation (3.27) |
| $\bar{L}$ | factor in the lift-to-thrust ratio; equation (5.6) |
| $M_{x'}, M_{y'}, M_{z'}$ | components of the (period averaged) hydrodynamic moment, referred to the origin of $C'$; equations (3.30)–(3.32) |
| $\bar{M}_{z',\mathrm{ref}}$ | factor in the pitching-moment-to-thrust ratio; equation (5.11) |
| $\bar{M}_{z',\mathrm{ref}}^{\pm}$ | maximum (+) and minimum (−) of $\bar{M}_{z',\mathrm{ref}}$ with respect to $\phi_\theta$ and $\hat{\theta}_t$ |
| $m_{x'}$ | rolling moment per unit length of the swimmer about the $x'$-axis; equation (3.15) |
| $\mathbf{N}, \mathbf{n}$ | normal to the left-side of the body, facing left; $\mathbf{n} = \mathbf{N}/|\mathbf{N}|$; equation (A 2) |
| $\mathbf{n}_0, \mathbf{n}_1$ | constituents of $\mathbf{n}$ associated with translational and rotational motions; equations (A 7) and (A 8) |

(*Continued.*)

| | |
|---|---|
| $\mathbf{n}_{\pm}$ | unit normal to dorsal and ventral edges of the swimmer; equation (A 9) |
| $P$ | (period averaged) power required to sustain the propulsion waves; equation (3.29) |
| $p$ | pressure |
| $\Delta p$ | pressure jump across the body of the swimmer (left minus right); equation (2.31) |
| Re | Reynolds number, Re $= vl/\nu$ |
| $S_w$ | wetted area of the swimmer |
| $s$ | local semi-span: half the distance between the dorsal and ventral edges |
| $T$ | thrust: the component of the (period averaged) hydrodynamic force along $-\mathbf{e}_x = -\mathbf{e}_{x'}$; equation (3.26) |
| $\bar{T}$ | factor in $T$; equation (5.2) |
| $\mathbf{T}_{\pm}$ | tangent to the dorsal and ventral edges of the body; equation (A 10) |
| $t$ | time |
| $w$ | local velocity of the swimmer's body relative to quiescent fluid, taken with the negative sign; normal-to-the-body component only; equation (C 3) |
| $w_0, w_1$ | constituents of $w$ associated with translational and rotational motions; equations (2.17) and (2.18) |
| $X_1, X_2$ | integral operators; equations (5.8)–(5.9) |
| $x, y, z$ | coordinates of a point relative to C |
| $x', y', z'$ | coordinates of a point relative to C′; equations (2.4)–(2.6) |
| $x_{cL}, x_{cm}, x_{cb}$ | coordinates of the centers of lift, mass and buoyancy |
| $x_n, x_t$ | coordinates of the cranial and caudal ends ('nose' and 'tail') in C and C′ alike; $x_n = 0$ and $x_t = 1$ by assumption |
| $y_0, \hat{y}_0$ | coordinate of the body centreline relative to C′; $\hat{y}_0$ is independent of time |
| $\hat{y}_t$ | in-plane deflection of the body centreline at the tail section; $\hat{y}_t = \hat{y}_0(x_t)$ |
| $Z$ | side-force: the component of the (period averaged) hydrodynamic force along $\mathbf{e}_{z'}$; equation (3.28) |
| $z_0$ | coordinate of the body centreline relative to C′ |
| $\hat{z}_0$ | modulating amplitude of the lateral propulsion waves; equation (4.1) |
| $z_b, z'_b$ | coordinate of the body surface relative to C′; equations (2.9) and (2.10) |
| $\hat{z}_t$ | amplitude of the lateral propulsion waves at the tail section; $\hat{z}_t = \hat{z}_0(x_t)$ |
| $\beta$ | ratio between submerged weight and buoyancy |
| $\zeta$ | (invariably) an integration variable |
| $\eta$ | propulsion efficiency; equation (3.41) |
| $\theta$ | twist angle |
| $\hat{\theta}_0$ | modulating amplitude of the torsional propulsion waves; equation (4.2) |
| $\theta_{J_1=0}$ | first non-trivial solution of $J_1(2x) = 0$, approximately 110° |
| $\hat{\theta}_t$ | amplitude of the torsional propulsion waves at the tail section; $\hat{\theta}_t = \hat{\theta}_0(x_t)$ |
| $\theta^{\pm}$ | $\hat{\theta}_t$ that maximizes (+) or minimizes (−) the pitching-moment-to-thrust ratio |
| $\iota$ | power per unit length; equation (3.20) |
| $\kappa$ | angular wavenumber; equations (4.1)–(4.2) |
| $\mu$ | potential jump across the body of the swimmer (left minus right); equation (2.12) |
| $\mu_n$ | nth moment of $\mu$; equation (2.20) |
| $\bar{\mu}_n$ | numerical factor in $\mu_n$; equations (2.29)–(2.30) |
| $\Pi_n$ | nth moment of pressure; equation (2.32) |
| $\hat{\sigma}_t$ | ratio of the maximal semi-span to the amplitude of the lateral waves at the tail section; $\hat{\sigma}_t = s_t/\hat{z}_t$ |

(*Continued.*)

**Table 1.** (Continued.)

| | |
|---|---|
| $\phi$ | perturbation velocity potential; equation (2.11) |
| $\phi_M$ | $\phi_\theta$ that maximizes or minimizes the pitching moment; equation (5.13) |
| $\phi_z$, $\phi_\theta$ | phases of the lateral and torsional propulsion waves at $t = x = 0$; equations (4.1)–(4.2) |
| $\psi_z$, $\psi_\theta$ | instantaneous phases of the lateral and torsional propulsion waves; equation (4.4) |
| $\omega$ | angular frequency; equations (4.1)–(4.2) |
| special symbols | |
| $\widehat{\dots}$ | typically, an amplitude |
| $\overline{\dots}$ | typically, a factor in the quantity bearing the same name |
| $\dot{\dots}$ | derivative with respect to a single argument |
| $\dots'$ | point function which is explicitly based on coordinates of the point in $C'$ |
| $\dots_0$ | typically, pertaining to the centreline |
| $\dots_{\mathrm{ref}}$ or $\dots_{,\mathrm{ref}}$ | pertaining to or referred to the reference section |
| $\dots_t$ or $\dots_{,t}$ | pertaining to or referred to the tail (caudal) section |
| $\langle\dots\rangle$ | average over a single period |
| $D/Dt$ | linearized Lagrangian derivative, $\partial/\partial t + \partial/\partial x$ |

The local semi-span (figure 1) is described by a monotonically increasing function $s: (x_n, x_t) \to (0, s_t)$; it is understood that $s(x_t) = s_t$. The body of the swimmer is assumed to be pronouncedly elongated, so both $s_t$ and $\max\limits_{x \in (x_n, x_t)}(\mathrm{d}s/\mathrm{d}x)$ are small as compared with unity.

The swimmer is allowed to bend in-plane (as if by arching its back—figure 1$e$), bend out of plane (as any anguilliform swimmer would—figure 1$c$), and twist about its central (cranio-caudal) axis (figure 1$d$). The surface of the swimmer will be parametrized either by

$$z' = z'_b(t, x, y'), \tag{2.7}$$

or by

$$z' = z_b(t, x, y). \tag{2.8}$$

It is assumed that each cross section of the swimmer does not deform during swimming and does not leave its respective $y'$–$z'$ plane. In this case, $y'$ and $y$ are related by (2.5) with $z = 0$, whereas $z'_b$ and $z_b$ can be expressed as

$$z'_b(t, x, y') = z_0(t, x) + (y' - y_0(t, x))\tan\theta(t, x) \tag{2.9}$$

and

$$z_b(t, x, y) = z_0(t, x) + y\sin\theta(t, x). \tag{2.10}$$

Deformations of the body are assumed small, so that $\max\limits_{t,x,y}|z_b(t, x, y)|$, $\max\limits_{t,x,y}|\partial z_b(t, x, y)/\partial t|$ and $\max\limits_{t,x,y}|\partial z_b(t, x, y)/\partial x|$ are small as compared with unity. Additional constraint on the allowed deformations will be introduced in the next section.

## 2.3. Underlying assumptions

As mentioned already in the Introduction, the plan is to find forces acting on the swimmer in the framework of the slender body theory. Its fundamentals can be found in quite a few references (e.g. [3–7,9,10]) and hence will not be repeated here; its main assumptions are recapitulated below.

Apart from the obvious assumptions on slenderness of the swimmer and smallness of its deformations (that were already made in the preceding section), the slender body theory relies on three basic assumptions: (i) the vortical regions in the flow are confined to the boundary layer on

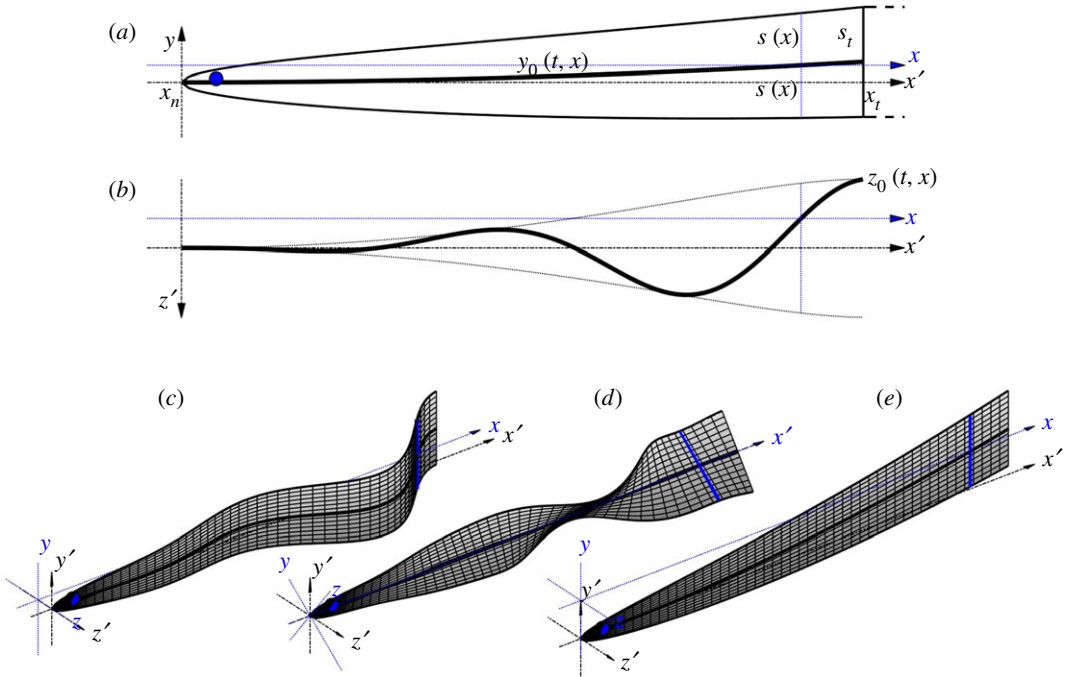

**Figure 1.** The model swimmer and the coordinate systems. Side ($a$) and top ($b$) views of an untwisted swimmer; axonometric projections of lateral and torsional waves are shown on ($c$) and ($d$); an exaggerated in-plane bend is shown on ($e$). The axes of $C'$ are shown with dash-dot lines. Shifted and rotated axes of $C$ are shown by blue dotted lines for a cross section that is marked by a thick blue line.

the surface of the swimmer and the wake behind it, (ii) the flow separates from (and only from) trailing edges of the body, and (iii) both the boundary layer and the part of the wake in the immediate proximity of the body are (vanishingly) thin. Under these assumptions, the velocity and pressure fields in the exterior of the boundary layer and the wake—and, in particular, on their outer boundary—can be found without finding them in interior of these regions. By associating thrust and power with the normal stresses on the surface of the body and the drag with the shear stresses [8, §3.1], the lack of knowledge of the flow field in the interior of the boundary layer still allows estimating thrust and power, but it disallows estimating drag. Shear stresses are hardly affected by undulations of the body [2], and since drag of an undeformed body can be estimated with a fairly good accuracy by empirical methods [11], lack of knowledge of its exact value should not affect the conclusions of this paper.

Location of the trailing edges along the body—these are the edges where the wake forms—has huge effect on the complexity of the solution for the forces acting on it. Having parts of the body embedded in the wake makes the solution unwieldly [5–8]; having the wake form at different parts of an edge during a tail-beat makes it intractable. Both cases are avoided here by ending the model swimmer at its widest section (see above), and limiting its admissible deformations to those for which the wake forms at (and only at) the widest section (appendix B).

## 2.4. Potential jump and its moments

The slender body theory furnishes the velocity and pressure fields in the (irrotational) exterior of the boundary layer and the wake as leading terms of the respective asymptotic series in the slenderness parameter—the ratio between typical lateral and longitudinal dimensions of the body. For the problem at hand, it can be the largest of the spatial derivatives of $y_0$, $z_0$ and $s\theta$. This theory can be derived formally, based on the method of matched asymptotic expansions [5,6,9,10], and informally, based on momentum considerations [3,7]. In the leading order with respect to the slenderness parameter all formulations are practically equivalent, and reduce the problem of finding the velocity and pressure fields near the body of the swimmer to that of finding a certain scalar field $\phi$ that

satisfies two-dimensional Laplace equation in every transverse plane crossing the body (i.e. at every $x \in (x_n, x_t)$), satisfies an impermeability condition on its surface, and vanishes at infinity.[3]

In the present case, the body of the swimmer in the transverse plane occupies the interior of the slit $\{(y, z) : y \in (-s(x), s(x)), z = 0\}$, and the general solution of the two-dimensional Laplace equation in its (unbounded) exterior is[4]

$$\phi(t, x, y, z) = \frac{1}{2\pi} \int_{-s(x)}^{s(x)} \frac{\mu(t, x, \zeta) z \, d\zeta}{(y - \zeta)^2 + z^2}, \tag{2.11}$$

where

$$\mu(t, x, y) = \phi(t, x, y, +0) - \phi(t, x, y, -0) \tag{2.12}$$

is the potential jump across the slit.[5] This general solution is yet to satisfy the impermeability condition on the surface of the slit,

$$\lim_{z \to \pm 0} \frac{\partial \phi(t, x, y, z)}{\partial z} = -w(t, x, y) \ \text{ for each } y \in (-s(x), s(x)) \tag{2.13}$$

(the right-hand side will be explicated shortly below), and the conjunction of (2.11) and (2.13) furnishes[6] an integro-differential equation for $\mu(t, x, \cdot)$,

$$\frac{1}{2\pi} \int_{-s(x)}^{s(x)} \frac{\partial \mu(t, x, \zeta)}{\partial \zeta} \frac{d\zeta}{y - \zeta} = w(t, x, y) \ \text{ for each } y \in (-s(x), s(x)). \tag{2.14}$$

The bar across the integral sign indicates principle value in Cauchy sense. Its solution,

$$\frac{\partial \mu(t, x, y)}{\partial y} = -\frac{2}{\pi} \frac{1}{\sqrt{s^2(x) - y^2}} \int_{-s(x)}^{s(x)} \sqrt{s^2(x) - \zeta^2} \frac{w(t, x, \zeta) \, d\zeta}{y - \zeta}, \tag{2.15}$$

immediately follows by the Söhngen inversion [9].[7] In the last three equations,

$$w(t, x, y) = w_0(t, x) + y w_1(t, x), \tag{2.16}$$

$$w_0(t, x) = -\cos \theta(t, x) \frac{Dz_0(t, x)}{Dt} + \sin \theta(t, x) \frac{Dy_0(t, x)}{Dt}, \tag{2.17}$$

$$w_1(t, x) = -\frac{D\theta(t, x)}{Dt}, \tag{2.18}$$

and $D/Dt$ stands for the linearized Lagrangian derivative, $\partial/\partial t + \partial/\partial x$; details can be found in appendix C.

Subject to

$$\mu(t, x, \pm s(x)) = 0, \tag{2.19}$$

equation (2.15) can be integrated on $(-s(x), y)$ to obtain $\mu(t, x, y)$, which can be substituted back in (2.11) to complete the solution for $\phi(t, x, y, z)$. This detailed solution will not be needed, however, and the first few moments of $\partial \mu / \partial y$,

$$\mu_n(t, x) = \frac{1}{s^{n+1}(x)} \int_{-s(x)}^{s(x)} \frac{\partial \mu(t, x, y)}{\partial y} y^n \, dy, \tag{2.20}$$

will suffice to obtain all relevant hydrodynamic forces and moments.

Thus,

$$\mu_n(t, x) = -\frac{2}{\pi s^{n+1}(x)} \int_{-s(x)}^{s(x)} \sqrt{s^2(x) - \zeta^2} w(t, x, \zeta) \, d\zeta \int_{-s(x)}^{s(x)} \frac{1}{\sqrt{s^2(x) - y^2}} \frac{y^n dy}{y - \zeta} \tag{2.21}$$

---

[3]For a body of zero thickness, $\phi$ can be identified with perturbation velocity potential, so that $\mathbf{v}' = \mathbf{e}_{x'} + \nabla' \phi$.

[4]Effectively, this is the potential of a distribution of doublets of intensity $\mu$ oriented along the $z$-axis and located in the interior of the slit [12].

[5]It can be verified by showing that $\lim_{z \to \pm 0} \phi(t, x, y, z) = \pm \mu(t, x, y)/2$.

[6]The simplest way to obtain the expression on the left is to integrate (2.11) by parts, differentiate second, and carefully compute the limit. Passage to the limit introduces the principal value to the integral. In this form, it appears in practically any textbook on aerodynamics [9,12].

[7]Having assumed that the flow does not separate anteriad of the widest point, implies that the flow turns round the edges. The limiting procedure of taking the thickness of the swimmer to zero introduces a square-root singularity in the velocity jump, $\partial \mu(t, x, y)/\partial y$. It does not introduce a discontinuity in the velocity potential, however, and hence (2.19).

by (2.15), and hence

$$\mu_n(t,x) = -\frac{2}{\pi s^{n+1}(x)} \int_{-s(x)}^{s(x)} \frac{y^n \mathrm{d}y}{\sqrt{s^2(x) - y^2}} \int_{-s(x)}^{s(x)} \frac{\sqrt{s^2(x) - \zeta^2}}{y - \zeta} \left(w_0(t,x) + \zeta w_1(t,x)\right) \mathrm{d}\zeta \tag{2.22}$$

by (2.16). Substituting $y = -s(x)\cos\theta_y$ and $\zeta = -s(x)\cos\theta_\zeta$, it becomes a combination

$$\mu_n(t,x) = -\frac{2}{\pi} \int_0^\pi (-1)^n \cos^n \theta_y \mathrm{d}\theta_y \int_0^\pi \left(w_0(t,x) - s(x)\cos\theta_\zeta w_1(t,x)\right) \frac{\sin^2\theta_\zeta \mathrm{d}\theta_\zeta}{\cos\theta_\zeta - \cos\theta_y} \tag{2.23}$$

of standard (Glauert) integrals [13], which yields

$$\mu_n(t,x) = (-1)^n \left(2w_0(t,x)C_{n+1} + s(x)w_1(t,x)(C_n - 2C_{n+2})\right), \tag{2.24}$$

where

$$C_n = \int_0^\pi \cos^n \theta_y \mathrm{d}\theta_y. \tag{2.25}$$

Among these, $C_{2n-1} = 0$ with any $n > 0$ by symmetry considerations, whereas

$$C_{2n} = \frac{1}{2^{2n}} \int_0^\pi (e^{i\theta} + e^{-i\theta})^{2n} \mathrm{d}\theta = \frac{\pi(2n)!}{2^{2n}(n!)^2}. \tag{2.26}$$

Consequently, (2.24) can be recast as

$$\mu_{2n}(t,x) = -2\pi s(x)w_1(t,x)\bar{\mu}_{2n} \tag{2.27}$$

and

$$\mu_{2n-1}(t,x) = -2\pi w_0(t,x)\bar{\mu}_{2n-1}, \tag{2.28}$$

where

$$\bar{\mu}_{2n}(t,x) = -\frac{1}{2}\left(\frac{(2n)!}{2^{2n}(n!)^2} - \frac{(2n+2)!}{2^{2n+1}((n+1)!)^2}\right) = \frac{(2n)!n}{2^{2n+1}(n+1)!n!} \tag{2.29}$$

and

$$\bar{\mu}_{2n-1} = \frac{(2n)!}{2^{2n}(n!)^2} \tag{2.30}$$

are certain numerical coefficients, and the convention $0! = 1$ applies. In particular, $\bar{\mu}_0 = 0$, $\bar{\mu}_1 = 1/2$, $\bar{\mu}_2 = 1/8$ and $\bar{\mu}_3 = 3/8$.

## 2.5. Pressure jump and its moments

The pressure jump across the body, $\Delta p(t, x, y) = p(t, x, y, +0) - p(t, x, y, -0)$, is given by

$$\Delta p(t,x,y) = -\frac{\mathrm{D}\mu(t,x,y)}{\mathrm{D}t} + \frac{\partial\mu(t,x,y)}{\partial y}\left(\sin\theta(t,x)\frac{\mathrm{D}z_0(t,x)}{\mathrm{D}t} + \cos\theta(t,x)\frac{\mathrm{D}y_0(t,x)}{\mathrm{D}t}\right) + \dots, \tag{2.31}$$

where the ellipsis comes to emphasize that the expression is correct only in the leading order with respect to the spatial derivatives of $z_0$, $y_0$ and $\theta$. Derivation of (2.31) can be found in appendix D. With (2.31) and (2.20), the $n$th-order pressure moment

$$\Pi_n(t,x) = -\int_{-s(x)}^{s(x)} \Delta p(t,x,y)y^n \, \mathrm{d}y, \tag{2.32}$$

becomes a combination

$$\Pi_n(t,x) = -\frac{1}{n+1}\frac{\mathrm{D}}{\mathrm{D}t}\left(s^{n+2}(x)\mu_{n+1}(t,x)\right) - s^{n+1}(x)\left(\sin\theta(t,x)\frac{\mathrm{D}z_0(t,x)}{\mathrm{D}t} + \cos\theta(t,x)\frac{\mathrm{D}y_0(t,x)}{\mathrm{D}t}\right)\mu_n(t,x) \tag{2.33}$$

of the respective moments of the potential jump, $\mu_n$ and $\mu_{n+1}$. Derivation of (2.33), as well as explicit expressions, relating $\Pi_n$ with $w_0$ and $w_1$, can be found in appendix E. The zeroth- and first-order pressure moments,

$$\Pi_0(t,x) = \pi\frac{\mathrm{D}}{\mathrm{D}t}\left(s^2(x)w_0(t,x)\right) \tag{2.34}$$

and

$$\Pi_1(t,x) = \frac{\pi}{8}\frac{D}{Dt}\left(s^4(x)w_1(t,x)\right) + \pi s^2(x)\left(\sin\theta(t,x)\frac{Dz_0(t,x)}{Dt} + \cos\theta(t,x)\frac{Dy_0(t,x)}{Dt}\right)w_0(t,x), \qquad (2.35)$$

are shown here because they will be actively used below. Note that the zeroth-order moment (the force per unit length) is independent of the shape of the body's centreline.

## 2.6. Leading-edge suction

When the thickness of the body tends to zero, the pressure at its leading edges becomes singular. The product of the body thickness and pressure remains finite, however, giving rise to what is known as the 'leading-edge suction'—the force acting on the edge and oriented along the normal to it (i.e. along $\mathbf{n}_\pm$—see appendix A). There are two leading edges, and the force (per unit length) acting on each one of them is

$$f_\pm(t,x) = \frac{\pi}{4}A_\pm^2(t,x), \qquad (2.36)$$

where

$$A_\pm(t,x) = \lim_{y\to\pm s(x)}\sqrt{s(x)\mp y}\frac{\partial\mu(t,x,y)}{\partial y} \qquad (2.37)$$

is the coefficient with the square-root singularity of $\mu$ at the respective edge [14]. Its explicit form is

$$\begin{aligned} A_\pm(t,x) &= \mp\frac{2}{\pi}\frac{1}{\sqrt{2s(x)}}\int_{-s(x)}^{s(x)}\sqrt{\frac{s(x)\pm\zeta}{s(x)\mp\zeta}}w(t,x,\zeta)\,d\zeta \\ &= \sqrt{2s(x)}\left(\mp w_0(t,x) - \frac{1}{2}w_1(t,x)s(x)\right) \end{aligned} \qquad (2.38)$$

by (2.15) and (2.16). The leading-edge suction,

$$f_\pm(t,x) = \frac{\pi}{2}s(x)\left(w_0(t,x) \pm \frac{1}{2}w_1(t,x)s(x)\right)^2, \qquad (2.39)$$

follows (2.38) by (2.36).

# 3. Forces and moments

## 3.1. Forces

The force per unit length acting on the body is given by

$$\mathbf{f}(t,x) = -\int_{-s(x)}^{s(x)}\Delta p(t,x,y)\mathbf{n}(t,x,y)dy + f_-(t,x)\mathbf{n}_-(t,x) + f_+(t,x)\mathbf{n}_+(t,x), \qquad (3.1)$$

or, what is equivalent by (A 6) and (2.32),

$$\mathbf{f}(t,x) = \Pi_0(t,x)\mathbf{n}_0(t,x) + \Pi_1(t,x)\mathbf{n}_1(t,x) + f_-(t,x)\mathbf{n}_-(t,x) + f_+(t,x)\mathbf{n}_+(t,x). \qquad (3.2)$$

Substituting (A 7), (A 8) and (A 11) from appendix A for the normal vectors, the three components of $\mathbf{f}(t,x)$ in C' become

$$f_{x'} = -\Pi_0\left(\cos\theta\frac{\partial z_0}{\partial x} - \sin\theta\frac{\partial y_0}{\partial x}\right) - \Pi_1\frac{\partial\theta}{\partial x} - (f_- + f_+)\frac{ds}{dx} - (f_+ - f_-)\left(\sin\theta\frac{\partial z_0}{\partial x} + \cos\theta\frac{\partial y_0}{\partial x}\right), \qquad (3.3)$$

$$f_{y'} = -\Pi_0\sin\theta + (f_+ - f_-)\cos\theta \qquad (3.4)$$

and

$$f_{z'} = +\Pi_0\cos\theta + (f_+ - f_-)\sin\theta; \qquad (3.5)$$

the arguments of all functions have been omitted for brevity. With (2.39), (2.34), (2.35), (2.17) and (2.18),

they yield

$$f_{x'} = -\pi\left(\cos\theta\frac{\partial z_0}{\partial x} - \sin\theta\frac{\partial y_0}{\partial x}\right)\frac{D}{Dt}\left(s^2 w_0\right) - \pi s^2 w_0\frac{\partial\theta}{\partial x}\left(\sin\theta\frac{Dz_0}{Dt} + \cos\theta\frac{Dy_0}{Dt}\right)$$
$$-\frac{\pi}{8}\frac{\partial\theta}{\partial x}\frac{D}{Dt}\left(s^4 w_1\right) - \left(\frac{\pi}{2}w_0^2\frac{\partial s^2}{\partial x} + \frac{\pi}{16}w_1^2\frac{\partial s^4}{\partial x}\right) - \pi s^2 w_0 w_1\left(\sin\theta\frac{\partial z_0}{\partial x} + \cos\theta\frac{\partial y_0}{\partial x}\right), \quad (3.6)$$

$$f_{y'} = -\pi\sin\theta\frac{D}{Dt}\left(s^2 w_0\right) + \pi s^2 w_0 w_1\cos\theta = -\pi\frac{D}{Dt}\left(\sin\theta\, s^2 w_0\right) \quad (3.7)$$

and
$$f_{z'} = \pi\cos\theta\frac{D}{Dt}\left(s^2 w_0\right) + \pi s^2 w_0 w_1\sin\theta = \pi\frac{D}{Dt}\left(\cos\theta\, s^2 w_0\right). \quad (3.8)$$

Exploiting (2.18) and (2.17), about half a page of algebraic manipulations on (3.6) (detailed in appendix F) furnishes

$$f_{x'} = \pi\frac{D}{Dt}\left(s^2 w_0\left(-\cos\theta\frac{\partial z_0}{\partial x} + \sin\theta\frac{\partial y_0}{\partial x}\right) - \frac{1}{8}s^4 w_1\frac{\partial\theta}{\partial x}\right) - \frac{\pi}{2}\frac{\partial}{\partial x}\left(s^2 w_0^2 + \frac{s^4}{8}w_1^2\right) \quad (3.9)$$

in a short form, and

$$f_{x'} = \pi\frac{D}{Dt}\left(s^2\left(\cos\theta\frac{Dz_0}{Dt} - \sin\theta\frac{Dy_0}{Dt}\right)\left(\cos\theta\frac{\partial z_0}{\partial x} - \sin\theta\frac{\partial y_0}{\partial x}\right) + \frac{1}{8}s^4\frac{D\theta}{Dt}\frac{\partial\theta}{\partial x}\right)$$
$$-\frac{\pi}{2}\frac{\partial}{\partial x}\left(s^2\left(\cos\theta\frac{Dz_0}{Dt} - \sin\theta\frac{Dy_0}{Dt}\right)^2 + \frac{s^4}{8}\left(\frac{D\theta}{Dt}\right)^2\right) \quad (3.10)$$

in a long one. Averaging (3.7)–(3.10) over a tail-beat period ($t_p$) yields

$$\langle f_{x'}\rangle = \frac{\pi}{2}\frac{\partial}{\partial x}\left\langle -s^2 w_0\left(2\cos\theta\frac{\partial z_0}{\partial x} - 2\sin\theta\frac{\partial y_0}{\partial x} + w_0\right) - \frac{1}{8}s^4 w_1\left(2\frac{\partial\theta}{\partial x} + w_1\right)\right\rangle \quad (3.11)$$

$$= \frac{\pi}{2}\frac{\partial}{\partial x}\left\langle s^2\cos^2\theta\left(\left(\frac{\partial z_0}{\partial x}\right)^2 - \left(\frac{\partial z_0}{\partial t}\right)^2\right)\right\rangle + \frac{\pi}{2}\frac{\partial}{\partial x}\left\langle s^2\sin^2\theta\left(\left(\frac{\partial y_0}{\partial x}\right)^2 - \left(\frac{\partial y_0}{\partial t}\right)^2\right)\right\rangle$$
$$-\frac{\pi}{2}\frac{\partial}{\partial x}\left\langle s^2\sin 2\theta\left(\frac{\partial z_0}{\partial x}\frac{\partial y_0}{\partial x} - \frac{\partial z_0}{\partial t}\frac{\partial y_0}{\partial t}\right)\right\rangle + \frac{\pi}{16}\frac{\partial}{\partial x}\left\langle s^4\left(\left(\frac{\partial\theta}{\partial x}\right)^2 - \left(\frac{\partial\theta}{\partial t}\right)^2\right)\right\rangle, \quad (3.12)$$

$$\langle f_{y'}\rangle = -\pi\frac{\partial}{\partial x}\left\langle s^2\sin\theta\, w_0\right\rangle = \pi\frac{\partial}{\partial x}\left\langle s^2\sin\theta\cos\theta\frac{Dz_0}{Dt} - s^2\sin^2\theta\frac{Dy_0}{Dt}\right\rangle \quad (3.13)$$

and
$$\langle f_{z'}\rangle = \pi\frac{\partial}{\partial x}\left\langle s^2\cos\theta\, w_0\right\rangle = \pi\frac{\partial}{\partial x}\left\langle -s^2\cos^2\theta\frac{Dz_0}{Dt} + s^2\sin\theta\cos\theta\frac{Dy_0}{Dt}\right\rangle, \quad (3.14)$$

where the angular brackets stand for the averaging operator, $\langle\ldots\rangle = (1/t_p)\int_0^{t_p}\ldots(t)\,dt$.

## 3.2. Rolling moment

The rolling moment (per unit length) about the $x'$-axis is given by

$$m_{x'}(t,x) = -\int_{-s(x)}^{s(x)}\Delta p(t,x,y)y\,dy + f_{z'}(t,x)y_0(t,x) - f_{y'}(t,x)z_0(t,x), \quad (3.15)$$

or, what is equivalent by (2.32),

$$m_{x'}(t,x) = \Pi_1(t,x) + f_{z'}(t,x)y_0(t,x) - f_{y'}(t,x)z_0(t,x). \quad (3.16)$$

Consistent with the direction of the $x$-axis, it is positive when rolling to the left. Introducing (2.35), (3.7) and (3.8), it yields

$$m_{x'} = \frac{\pi}{8}\frac{D}{Dt}\left(s^4 w_1\right) + \pi s^2\left(\sin\theta\frac{Dz_0}{Dt} + \cos\theta\frac{Dy_0}{Dt}\right)w_0 + \pi z_0\frac{D}{Dt}\left(\sin\theta\, s^2 w_0\right) + \pi y_0\frac{D}{Dt}\left(\cos\theta\, s^2 w_0\right), \quad (3.17)$$

where the arguments of the respective functions have been removed for brevity, as in (3.3). The second

term cancels out with a remainder of the union of the last two terms, leaving

$$
\begin{aligned}
m_{x'} &= \pi \frac{D}{Dt}\left(s^2\left(z_0 \sin\theta + y_0 \cos\theta\right)w_0 + \frac{1}{8}s^4 w_1\right)\\
&= \pi \frac{D}{Dt}\left(s^2\left(z_0 \sin\theta + y_0 \cos\theta\right)\left(-\cos\theta\frac{Dz_0}{Dt} + \sin\theta\frac{Dy_0}{Dt}\right) - \frac{1}{8}s^4\frac{D\theta}{Dt}\right)
\end{aligned}
\tag{3.18}
$$

by (2.17) and (2.18). Assuming $\theta$ to be periodic with zero mean, the tail-beat-average of (3.18) yields

$$
\begin{aligned}
\langle m_{x'}\rangle &= \frac{\pi}{2}\frac{\partial}{\partial x}\left\langle -s^2\left(y_0 + z_0\sin 2\theta + y_0\cos 2\theta\right)\frac{Dz_0}{Dt} + s^2\left(z_0 + y_0\sin 2\theta - z_0\cos 2\theta\right)\frac{Dy_0}{Dt}\right\rangle\\
&= \frac{\pi}{2}\frac{\partial}{\partial x}\left(s^2\left\langle z_0\frac{Dy_0}{Dt} - y_0\frac{Dz_0}{Dt} - \frac{D(y_0 z_0)}{Dt}\cos 2\theta + \frac{1}{2}\left(\frac{Dy_0^2}{Dt} - \frac{Dz_0^2}{Dt}\right)\sin 2\theta\right\rangle\right).
\end{aligned}
\tag{3.19}
$$

## 3.3. Power

The power (per unit length) needed to sustain the deformation waves is given by

$$
\begin{aligned}
\iota(t,x) &= \int_{-s(x)}^{s(x)} \Delta p(t,x,y)\left(\cos\theta(t,x)\frac{\partial z_0(t,x,y)}{\partial t} - \sin\theta(t,x)\frac{\partial y_0(t,x)}{\partial t} + y\frac{\partial\theta(t,x)}{\partial t}\right)dy\\
&\quad - \left(f_+(t,x) - f_-(t,x)\right)\left(\sin\theta(t,x)\frac{\partial z_0(t,x,y)}{\partial t} + \cos\theta(t,x)\frac{\partial y_0(t,x)}{\partial t}\right);
\end{aligned}
\tag{3.20}
$$

the factor with the pressure jump in the first term is the normal-to-the-surface component of the swimmer's velocity in C′; the factor with the leading-edge suction in the second term is the normal-to-the-edge component of the swimmer's velocity in the same reference frame. Thus,

$$
\begin{aligned}
\iota(t,x) &= -\left(\cos\theta(t,x)\frac{\partial z_0(t,x)}{\partial t} - \sin\theta(t,x)\frac{\partial y_0(t,x)}{\partial t}\right)\Pi_0(t,x) - \frac{\partial\theta(t,x)}{\partial t}\Pi_1(t,x)\\
&\quad - \pi s^2(x)w_0(t,x)w_1(t,x)\left(\sin\theta(t,x)\frac{\partial z_0(t,x,y)}{\partial t} + \cos\theta(t,x)\frac{\partial y_0(t,x)}{\partial t}\right)
\end{aligned}
\tag{3.21}
$$

by (2.32) and (2.39), and, consequently,

$$
\begin{aligned}
\iota &= -\pi\left(\cos\theta\frac{\partial z_0}{\partial t} - \sin\theta\frac{\partial y_0}{\partial t}\right)\frac{D}{Dt}\left(s^2 w_0\right) - \pi s^2 w_0\frac{\partial\theta}{\partial t}\left(\sin\theta\frac{Dz_0}{Dt} + \cos\theta\frac{Dy_0}{Dt}\right)\\
&\quad - \frac{\pi}{8}\frac{\partial\theta}{\partial t}\frac{D}{Dt}\left(s^4 w_1\right) - \pi s^2 w_0 w_1\left(\sin\theta\frac{\partial z_0}{\partial t} + \cos\theta\frac{\partial y_0}{\partial t}\right)
\end{aligned}
\tag{3.22}
$$

by (2.34) and (2.35) (or (E 3) and (E 4)); again, the arguments of the respective functions have been removed for brevity. With an intermediate step shown in appendix G,

$$
\iota = \pi\frac{D}{Dt}\left(s^2 w_0\left(-\cos\theta\frac{\partial z_0}{\partial t} + \sin\theta\frac{\partial y_0}{\partial t}\right) - \frac{1}{8}s^4 w_1\frac{\partial\theta}{\partial t}\right) - \frac{\pi}{2}\frac{\partial}{\partial t}\left(s^2 w_0^2 + \frac{1}{8}s^4 w_1^2\right)
\tag{3.23}
$$

by (2.17) and (2.18). Its tail-beat-average is

$$
\langle\iota\rangle = -\pi\frac{\partial}{\partial x}\left\langle s^2 w_0\left(\cos\theta\frac{\partial z_0}{\partial t} - \sin\theta\frac{\partial y_0}{\partial t}\right)\right\rangle - \frac{\pi}{8}\frac{\partial}{\partial x}\left\langle s^4 w_1\frac{\partial\theta}{\partial t}\right\rangle
\tag{3.24}
$$

(equation (2.18) was used in the last term); or, explicitly,

$$
\begin{aligned}
\langle\iota\rangle &= \pi\frac{\partial}{\partial x}\left\langle s^2\cos^2\theta\frac{Dz_0}{Dt}\frac{\partial z_0}{\partial t}\right\rangle + \pi\frac{\partial}{\partial x}\left\langle s^2\sin^2\theta\frac{Dy_0}{Dt}\frac{\partial y_0}{\partial t}\right\rangle + \frac{\pi}{8}\frac{\partial}{\partial x}\left\langle s^4\frac{\partial\theta}{\partial t}\frac{D\theta}{Dt}\right\rangle\\
&\quad - \pi\frac{\partial}{\partial x}\left\langle s^2\sin\theta\cos\theta\left(\frac{Dz_0}{Dt}\frac{\partial y_0}{\partial t} + \frac{\partial z_0}{\partial t}\frac{Dy_0}{Dt}\right)\right\rangle
\end{aligned}
\tag{3.25}
$$

by (2.17) and (2.18).

## 3.4. Integral quantities

Hydrodynamic forces and moments acting on the entire body follow the above by quadratures. Explicit expressions for the respective time-dependent quantities turn unwieldy, but can be found in full in the electronic supplementary material, S1. Their coherence was verified by comparison with numerical simulations based on the vortex lattice method [12]. The rest of this manuscript addresses time-averaged quantities only. In an attempt to make all relevant expressions as short as possible, $w_0$ and $w_1$ (equations (2.17) and (2.18)) are left here unexpanded. Thus,

$$T = -\int_{x_n}^{x_t} \langle f_{x'} \rangle \mathrm{d}x = \frac{\pi}{2} s_t^2 \left\langle w_0 \left( 2 \cos \theta \frac{\partial z_0}{\partial x} - 2 \sin \theta \frac{\partial y_0}{\partial x} + w_0 \right) + \frac{1}{8} s_t^2 w_1 \left( 2 \frac{\partial \theta}{\partial x} + w_1 \right) \right\rangle_{x=x_t} \tag{3.26}$$

is the effective thrust (it follows by (3.11)),

$$L = \int_{x_n}^{x_t} \langle f_{y'} \rangle \, \mathrm{d}x = -\pi s_t^2 \langle w_0 \sin \theta \rangle_{x=x_t} \tag{3.27}$$

is the lift (it follows by (3.13)),

$$Z = \int_{x_n}^{x_t} \langle f_{z'} \rangle \mathrm{d}x = \pi s_t^2 \langle w_0 \cos \theta \rangle_{x=x_t} \tag{3.28}$$

is the lateral force (it follows by (3.14)),

$$P = \int_{x_n}^{x_t} \langle \iota \rangle \, \mathrm{d}x = -\pi s_t^2 \left\langle w_0 \left( \cos \theta \frac{\partial z_0}{\partial t} - \sin \theta \frac{\partial y_0}{\partial t} \right) + \frac{1}{8} s_t^2 \frac{\partial \theta}{\partial t} w_1 \right\rangle_{x=x_t} \tag{3.29}$$

is power required to sustain the deformation waves (it follows by (3.24)),

$$M_{x'} = \int_{x_n}^{x_t} \langle m_{x'} \rangle \, \mathrm{d}x = \pi s_t^2 \left\langle w_0 \left( z_0 \sin \theta + y_0 \cos \theta \right) \right\rangle_{x=x_t} \tag{3.30}$$

is the rolling moment about the $x'$-axis (it follows by (3.18) and (3.19)), and finally,

$$M_{y',\mathrm{ref}} = -(x_t - x_{\mathrm{ref}})Z + M_{y',t} \tag{3.31}$$

and

$$M_{z',\mathrm{ref}} = (x_t - x_{\mathrm{ref}})L + M_{z',t} \tag{3.32}$$

are the yawing and pitching moments about some $x = x_{\mathrm{ref}}$, where

$$M_{y',t} = -\int_{x_n}^{x_t} \langle f_{z'}(\cdot,x) \rangle (x - x_t) \, \mathrm{d}x = \pi \int_{x_n}^{x_t} s^2(x) \left\langle \cos \theta(\cdot,x)\, w_0(\cdot,x) \right\rangle \mathrm{d}x \tag{3.33}$$

and

$$M_{z',t} = \int_{x_n}^{x_t} \langle f_{y'}(\cdot,x) \rangle (x - x_t) \, \mathrm{d}x = \pi \int_{x_n}^{x} s^2(x) \left\langle \sin \theta(\cdot,x)\, w_0(\cdot,x) \right\rangle \mathrm{d}x \tag{3.34}$$

are the respective moments about the tail section (they follow by (3.13) and (3.14)). Admittedly, the last four equations are approximations, because they tacitly neglect the moments due to thrust as compared with those due to lift and side force. To remain consistent with directions of the respective axes, the pitching and yawing moments are defined as positive when pushing the nose down and left. Thrust is defined positive when pushing forwards. Classical results of the slender body theory [3],

$$T = \frac{\pi}{2} s_t^2 \left\langle \left( \frac{\partial z_0}{\partial t} \right)^2 - \left( \frac{\partial z_0}{\partial x} \right)^2 \right\rangle_{x=x_t} \tag{3.35}$$

and

$$P = \pi s_t^2 \left\langle \frac{\partial z_0}{\partial t}\left(\frac{\partial z_0}{\partial t} + \frac{\partial z_0}{\partial x}\right)\right\rangle_{x=x_t}, \tag{3.36}$$

follow (3.26) and (3.29) with $\theta = 0$ by (2.17) and (2.18).

## 3.5. Hydrodynamic losses and propulsion efficiency

The difference, $\Delta P = P - T$, between the power used and the power made good (recall that in dimensionless units, the swimming speed is unity, and hence the power made good, which is the product of thrust and speed, equals thrust) is the power lost to the fluid. It can be computed from (3.29) and (3.26),

$$\Delta P = P - T = \pi s_t^2 \left(\frac{1}{2}\left\langle w_0^2(\cdot, x_t)\right\rangle + \frac{1}{16}s_t^2\left\langle w_1^2(\cdot, x_t)\right\rangle\right), \tag{3.37}$$

but it also can be computed directly from the rate at which the kinetic energy is added to the fluid at the tail section [15],

$$P_t = \frac{1}{2}\left\langle \int_{-s_t}^{s_t} \mu(\cdot, x_t, y) w(\cdot, x_t, y)\, dy\right\rangle. \tag{3.38}$$

Introducing (2.16)—and noting (2.19)—it can be integrated by parts to obtain

$$\begin{aligned}P_t &= -\frac{1}{2}\left\langle \int_{-s_t}^{s_t} \frac{\partial \mu(\cdot, x_t, y)}{\partial y}\left(w_0(\cdot, x_t)y + \frac{1}{2}w_1(\cdot, x_t)y^2\right) dy\right\rangle \\ &= -\frac{1}{2}s_t^2\left\langle w_0(\cdot, x_t)\mu_1(\cdot, x_t)\right\rangle - \frac{1}{4}s_t^3\left\langle w_1(\cdot, x_t)\mu_2(\cdot, x_t)\right\rangle,\end{aligned} \tag{3.39}$$

which, in turn, yields

$$P_t = \pi s_t^2 \left(\frac{1}{2}\left\langle w_0^2(\cdot, x_t)\right\rangle + \frac{1}{16}s_t^2\left\langle w_1^2(\cdot, x_t)\right\rangle\right) \tag{3.40}$$

by (2.27) and (2.28). Indeed, $P_t = \Delta P$.

The (hydrodynamic) propulsion efficiency,

$$\eta = \frac{T}{P} = 1 - \frac{\Delta P}{P}, \tag{3.41}$$

is commonly defined as the ratio of the power made good, $T$ (see the opening paragraph of this section) and the power spent, $P$. Introducing (3.29) and (3.26), it becomes

$$\eta = \frac{\left\langle w_0\left(2\cos\theta\frac{\partial z_0}{\partial x} - 2\sin\theta\frac{\partial y_0}{\partial x} + w_0\right) + \frac{1}{8}s_t^2 w_1\left(2\frac{\partial\theta}{\partial x} + w_1\right)\right\rangle_{x=x_t}}{\left\langle -w_0\left(2\cos\theta\frac{\partial z_0}{\partial t} - 2\sin\theta\frac{\partial y_0}{\partial t}\right) - \frac{1}{8}s_t^2 2\frac{\partial\theta}{\partial t}w_1\right\rangle_{x=x_t}}; \tag{3.42}$$

it is reminded that $w_0$ and $w_1$ are given by (2.17) and (2.18).

# 4. Harmonic waves

## 4.1. Basic expressions

Based on Graham *et al.* [1], deformations of a swimming snake appear as a combination of lateral,

$$z_0(t, x) = \hat{z}_0(x)\cos(\omega t - \kappa x + \phi_z), \tag{4.1}$$

and torsional,

$$\theta(t, x) = \hat{\theta}_0(x)\cos(\omega t - \kappa x + \phi_\theta), \tag{4.2}$$

harmonic waves, and time-independent flex in the $x'$–$y'$ plane,

$$y_0(t, x) = \hat{y}_0(x). \tag{4.3}$$

Here, $\omega$ and $\kappa$ are the (angular) frequency and the (angular) wavenumber, $\phi_z$ and $\phi_\theta$ are the phase angles, $\hat{z}_0$ and $\hat{\theta}_0$ are the modulating amplitudes. $\hat{z}_0$ and $\hat{\theta}_0$ are assumed continuous, monotonic and non-negative on $(x_n, x_t)$, $d\hat{y}_0/dx$ is assumed continuous on $(x_n, x_t)$. The arguments of the cosines in (4.1) and (4.2) will be abbreviated by

$$\psi_z(t,x) = \omega t - \kappa x + \phi_z, \quad \psi_\theta(t,x) = \omega t - \kappa x + \phi_\theta \tag{4.4}$$

below, and, without a loss of generality, $\phi_z$ will be set zero. It is noted that

$$\frac{\partial z_0(t,x)}{\partial t} = -\omega \hat{z}_0(x) \sin \psi_z(t,x) \tag{4.5}$$

and

$$\frac{\partial z_0(t,x)}{\partial x} = \kappa \hat{z}_0(x) \sin \psi_z(t,x) + \dot{\hat{z}}_0(x) \cos \psi_z(t,x), \tag{4.6}$$

where an overdot marks the derivative of a function with respect to its single argument. Similar expressions hold for $\theta$.

Now, with any real $a$ and $\psi$, $\cos(a \cos \psi)$ and $\sin(a \cos \psi)$ can be expanded into the respective Fourier series

$$\cos(a \cos \psi) = J_0(a) + 2\sum_{m=1}^{\infty} (-1)^m J_{2m}(a) \cos 2m\psi \tag{4.7}$$

and

$$\sin(a \cos \psi) = -2\sum_{m=1}^{\infty} (-1)^m J_{2m-1}(a) \cos(2m-1)\psi, \tag{4.8}$$

where $J_0, J_1, \ldots$ are the Bessel functions of the first kind (these expansions are based on standard integrals appearing in eqns 3.715.13 and 3.715.18 of [16]). In particular,

$$2\cos^2\left(\theta(t,x)\right) = 1 + \cos\left(2\theta(t,x)\right)$$
$$= 1 + J_0\left(2\hat{\theta}_0(x)\right) + 2\sum_{m=1}^{\infty} (-1)^m J_{2m}\left(2\hat{\theta}_0(x)\right) \cos\left(2m\psi_\theta(t,x)\right), \tag{4.9}$$

and, similarly, $2\sin^2(\theta(t,x)) = 1 - \cos(2\theta(t,x))$ is given by a variant of (4.9) with minuses replacing the pluses. After introducing these in (3.12), (3.13), (3.14), (3.19) and (3.25)—or, rather, directly in (3.26)–(3.34)—a few pages of tedious (but rather straightforward) algebra yield

$$T = \frac{\pi s_t^2}{8}\left((\omega^2 - \kappa^2)\hat{z}_t^2 - \dot{\hat{z}}_t^2\right)\left(1 + J_0(2\hat{\theta}_t) + J_2(2\hat{\theta}_t)\cos 2\phi_\theta\right) + \frac{\pi s_t^4}{32}\left((\omega^2 - \kappa^2)\hat{\theta}_t^2 - \dot{\hat{\theta}}_t^2\right)$$
$$- \frac{\pi s_t^2}{4}\left(J_2(2\hat{\theta}_t)\dot{\hat{z}}_t(\kappa \hat{z}_t \sin 2\phi_\theta - \dot{\hat{z}}_t \cos 2\phi_\theta) + 2J_1(2\hat{\theta}_t)\dot{\hat{y}}_t(\kappa \hat{z}_t \sin \phi_\theta - \dot{\hat{z}}_t \cos \phi_\theta)\right)$$
$$- \frac{\pi s_t^2}{4}\left(1 - J_0(2\hat{\theta}_t)\right)\dot{\hat{y}}_t^2, \tag{4.10}$$

$$L = \frac{\pi s_t^2}{2} J_1(2\hat{\theta}_t)\left(\hat{z}_t(\omega - \kappa)\sin \phi_\theta + \dot{\hat{z}}_t \cos \phi_\theta\right) - \frac{\pi s_t^2}{2}\dot{\hat{y}}_t\left(1 - J_0(2\hat{\theta}_t)\right), \tag{4.11}$$

$$P = \frac{\pi s_t^2}{4}\omega(\omega - \kappa)\left(\left(1 + J_0(2\hat{\theta}_t) + J_2(2\hat{\theta}_t)\cos 2\phi_\theta\right)\hat{z}_t^2 + \frac{s_t^2 \hat{\theta}_t^2}{4}\right)$$
$$- \frac{\pi s_t^2}{4}\omega J_2(2\hat{\theta}_t)\hat{z}_t\dot{\hat{z}}_t \sin 2\phi_\theta - \frac{\pi s_t^2}{2}\omega J_1(2\hat{\theta}_t)\dot{\hat{y}}_t\hat{z}_t \sin \phi_\theta, \tag{4.12}$$

$$M_{z',t} = -\frac{\pi}{2}\int_{x_n}^{x_t} s^2(x)\left(\hat{z}_0(x)(\omega - \kappa)\sin \phi_\theta + \dot{\hat{z}}_0(x)\cos \phi_\theta\right)J_1\left(2\hat{\theta}_0(x)\right) dx$$
$$+ \frac{\pi}{2}\int_{x_n}^{x_t} s^2(x)\dot{\hat{y}}_0(x)\left(1 - J_0\left(2\hat{\theta}_0(x)\right)\right) dx \tag{4.13}$$

and also

$$Z = M_{x'} = M_{y',t} = 0. \tag{4.14}$$

The subscript '$t$' universally marks the values of the respective functions at the tail section: $\hat{y}_t = \hat{y}_0(x_t)$, $\hat{z}_t = \hat{z}_0(x_t)$, $\hat{\theta}_t = \hat{\theta}_0(x_t)$, and $s_t = s(x_t)$.

## 4.2. An extension

Extended variants of (4.10)–(4.14), which are based on

$$y_0(t, x) = \hat{y}_0(x) + \hat{v}_0(x) \cos(\omega t - \kappa x + \phi_y) \tag{4.15}$$

instead of (4.3), can be found in electronic supplementary material, S2. Deemed unwieldy to be used without simplifying them back to (4.10)–(4.14), they do offer an insight that two mutually perpendicular lateral harmonic waves do not interact in any of the tail-beat-averaged quantities appearing in (4.10)–(4.14), except for $M_{x'}$. This result could have been expected for lift, side force and the associated moments (that follow from (3.27) and (3.28)), but it could hardly be expected for thrust and power (that follow from (3.26) and (3.29)).

## 4.3. Limiting cases

The respective limits of (4.10)–(4.13)

$$\lim_{\hat{\theta}_t \to 0} T = \frac{\pi s_t^2}{4} \left( (\omega^2 - \kappa^2) \hat{z}_t^2 - \dot{\hat{z}}_t^2 \right), \tag{4.16}$$

$$\lim_{\hat{\theta}_t \to 0} L = 0, \tag{4.17}$$

$$\lim_{\hat{\theta}_t \to 0} P = \frac{\pi}{2} s_t^2 \omega(\omega - \kappa) \hat{z}_t^2 \tag{4.18}$$

and

$$\lim_{\hat{\theta}_t \to 0} M_{z',t} = 0, \tag{4.19}$$

recover the expressions of the classical elongated body theory [5,7,8]. Thrust can be obtained only when the phase velocity of the propulsion waves, $u = \omega/\kappa$, exceeds the swimming velocity.

The limits:

$$\lim_{\hat{z}_t \to 0} T = \frac{\pi s_t^4}{32} \left( (\omega^2 - \kappa^2) \hat{\theta}_t^2 - \dot{\hat{\theta}}_t^2 \right) - \frac{\pi s_t^2}{4} \left( 1 - J_0(2\hat{\theta}_t) \right) \dot{\hat{y}}_t^2, \tag{4.20}$$

$$\lim_{\hat{z}_t \to 0} L = -\frac{\pi}{2} s_t^2 \dot{\hat{y}}_t \left( 1 - J_0(2\hat{\theta}_t) \right), \tag{4.21}$$

$$\lim_{\hat{z}_t \to 0} P = \frac{\pi}{16} s_t^4 \omega(\omega - \kappa) \hat{\theta}_t^2 \tag{4.22}$$

and

$$\lim_{\hat{z}_t \to 0} M_{z',t} = \frac{\pi}{2} \int_{x_n}^{x_t} s^2(x) \dot{\hat{y}}_0(x) \left( 1 - J_0(2\hat{\theta}_0(x)) \right) \mathrm{d}x, \tag{4.23}$$

elucidate the equivalence between the classical anguilliform swimming gait based on backward-propagating lateral displacement waves, and its variant (or, rather, a variant of a gymnotiform gait [4]) based on backward-propagating torsional waves. In the context of propulsion, the effective amplitude of the edge displacement due to torsion, $s\hat{\theta}_0/\sqrt{8}$, is equivalent to the translation amplitude $\hat{z}_0$. Of course, twist interacts with the angle of attack (manifested in $\dot{\hat{y}}_0$), whereas translation alone does not, and hence additional terms proportional to powers of $\dot{\hat{y}}_0$ are found in (4.20), (4.21) and (4.23). In particular, the factor with $\dot{\hat{y}}_t$ in the expression for lift, $(\pi s_t^2/2)\left(1 - J_0(2\hat{\theta}_t)\right)$, can be identified with the lift slope coefficient $L_\alpha$ of a twisting swimmer,[8] whereas the ratio of the factor with $\dot{\hat{y}}_t^2$ in the expression for thrust and $L_\alpha^2$, can be identified with the induced drag coefficient.

The limits,

$$\lim_{\substack{\hat{z}_t \to 0 \\ \dot{\hat{\theta}}_t \to 0}} T = \frac{\pi s_t^2}{8} (\omega^2 - \kappa^2) \left( \left( 1 + J_0(2\hat{\theta}_t) + J_2(2\hat{\theta}_t) \cos 2\phi_\theta \right) \hat{z}_t^2 + \frac{s_t^2 \hat{\theta}_t^2}{4} \right)$$

$$- \frac{\pi s_t^2}{2} \kappa J_1(2\hat{\theta}_t) \dot{\hat{y}}_t \hat{z}_t \sin \phi_\theta - \frac{\pi s_t^2}{4} \left( 1 - J_0(2\hat{\theta}_t) \right) \dot{\hat{y}}_t^2, \tag{4.24}$$

$$\lim_{\hat{z}_t \to 0} L = \frac{\pi s_t^2}{2} J_1(2\hat{\theta}_t) \hat{z}_t (\omega - \kappa) \sin \phi_\theta - \frac{\pi s_t^2}{2} \dot{\hat{y}}_t \left( 1 - J_0(2\hat{\theta}_t) \right) \tag{4.25}$$

[8]Maximal list slope of $0.7\pi s_t^2$ is obtained at $\hat{\theta}_t \approx 110°$, where $J_1(2\theta_t) = 0$. It is 70% of what it would have been if the swimmer was swimming on its side.

and

$$\lim_{\dot{z}_t \to 0} P = \frac{\pi s_t^2}{4}\,\omega(\omega - \kappa)\left(\left(1 + J_0(2\hat{\theta}_t) + J_2(2\hat{\theta}_t)\cos 2\phi_\theta\right)\hat{z}_t^2 + \frac{s_t^2\,\hat{\theta}_t^2}{4}\right)$$
$$- \frac{\pi s_t^2}{2}\,\omega J_1(2\hat{\theta}_t)\dot{\hat{y}}_t\hat{z}_t\sin\phi_\theta \tag{4.26}$$

(equation (4.13) for the pitching moment remains unchanged), furnish a convenient framework for subsequent analysis. $\dot{\hat{z}}_t = 0$ and $\dot{\hat{\theta}}_t = 0$ reflect free-end conditions at $x = x_t$,[9] and have been actually observed with a swimming *H. platurus* [1] (figure 4).

Remarkably, a combination of (4.24)–(4.26),

$$\lim_{\substack{\dot{z}_t \to 0 \\ \dot{\theta}_t \to 0}} T = \lim_{\dot{z}_t \to 0}\left(\frac{\omega + \kappa}{2\omega}P + \frac{1}{2}\dot{\hat{y}}_t L\right), \tag{4.27}$$

suggests

$$\lim_{\substack{\dot{z}_t \to 0 \\ \dot{\theta}_t \to 0}} \eta = \lim_{\substack{\dot{z}_t \to 0 \\ \dot{\theta}_t \to 0}} \frac{T}{P} = \frac{\omega + \kappa}{2\omega} + \frac{1}{2}\dot{\hat{y}}_t \lim_{\dot{z}_t \to 0}\frac{L}{P} \tag{4.28}$$

by (3.41). In other words, generating lift with no angle of attack ($\dot{\hat{y}}_t = 0$) carries no energetic penalty. Generating positive lift when swimming tail high directs the normal-to-the-body component of the hydrodynamic force forwards and improves efficiency; conversely, generating positive lift by swimming tail-low worsens it. In fact, when $\phi_\theta = 0$, the last term in (4.24) (and hence in (4.28)) becomes identified with the induced drag.

The fact that thrust does not vanish when the power does, reflects a (possible) problem in separation of thrust and drag. Reclassifying thrust at zero power as drag, as was done in [17] for a bird in flapping flight, does not work here. As opposed to large birds, which can generate lift without flapping, slender flat swimmers cannot generate lift without twisting their bodies. Assuming $\dot{\hat{y}}_t \to 0$, in addition to $\dot{\hat{z}}_t \to 0$ and $\dot{\hat{\theta}}_t \to 0$, furnishes a fortuitous solution for the remaining part of the paper, but the general problem will have to be addressed somewhere.

# 5. Effects of torsion

## 5.1. Thrust

Equations (4.11) and (4.13) answer part of the question that started this study: an interaction between torsional and lateral waves can indeed generate lift and pitching moment. It remains to assess their magnitudes, and this is the subject matter of this section. To make the analysis concise, it will be limited to those cases where $\dot{\hat{z}}_t = \dot{\hat{\theta}}_t = \dot{\hat{y}}_t = 0$. The ubiquitous ratio $s_t/\hat{z}_t$ of the tail semi-span $s_t$ to the amplitude of the tail displacement $\hat{z}_t$ will be denoted here by $\hat{\sigma}_t$. It is a small quantity for a swimming sea-snake (it equals 0.11 for *H. platurus*—appendix I), but one can easily conceive a swimmer for which $\hat{\sigma}_t$ is of the order of unity.

In the case where $\dot{\hat{z}}_t = \dot{\hat{\theta}}_t = \dot{\hat{y}}_t = 0$, equation (4.10) can be rearranged as

$$T = \frac{\pi s_t^2}{4}\hat{z}_t^2(\omega^2 - \kappa^2)\bar{T}(\phi_\theta, \hat{\theta}_t, \hat{\sigma}_t), \tag{5.1}$$

where the factor

$$\bar{T}(\phi_\theta, \hat{\theta}_t, \hat{\sigma}_t) = \frac{1}{2}\left(1 + J_0(2\hat{\theta}_t) + J_2(2\hat{\theta}_t)\cos 2\phi_\theta\right) + \frac{1}{8}\hat{\sigma}_t^2\hat{\theta}_t^2 \tag{5.2}$$

manifests the effect of torsional waves on thrust—in fact, $\bar{T}(\phi_\theta, 0, \hat{\sigma}_t) = 1$, for any $\phi_\theta$ and any $\hat{\sigma}_t$. It is a (local) extremum of $\bar{T}$, which is a maximum when $\hat{\sigma}_t^2 < 4 - 2\cos 2\phi_\theta$ (figure 2a,c), and a minimum otherwise (figure 2d).[10] The minimum of $\bar{T}$ is invariably associated with $\phi_\theta = \pi/2$ and $\hat{\theta}_t$ ranging

---

[9]A swimming body can be modelled as a cantilever subjected to internal (muscles) and external (hydrodynamic) forces. There is no concentrated hydrodynamic force that acts on it at $x = x_t$ from the outside, and there are no muscles at $x = x_t$ to bend it from inside. The free-end condition for a cantilever is $\partial^2 z_0/\partial x^2 = 0$ at $x = x_t$, where $\partial^2 z_0/\partial x^2 = (\ddot{z}_0 - \kappa^2\hat{z}_0)\cos\psi_z + 2\kappa\dot{\hat{z}}_0\sin\psi_z$ by (4.1). A necessary condition to satisfy it at all times is $\dot{\hat{z}}_t = 0$. The same is true for $\hat{\theta}_t$.

[10]When $\hat{\theta}_t \ll 1$, $\bar{T}(\phi_\theta, \hat{\theta}_t, \hat{\sigma}_t) = 1 + (-4 + 2\cos 2\phi_\theta + \hat{\sigma}_t^2)(\hat{\theta}_t^2/8) + O(\hat{\theta}_t^4)$.

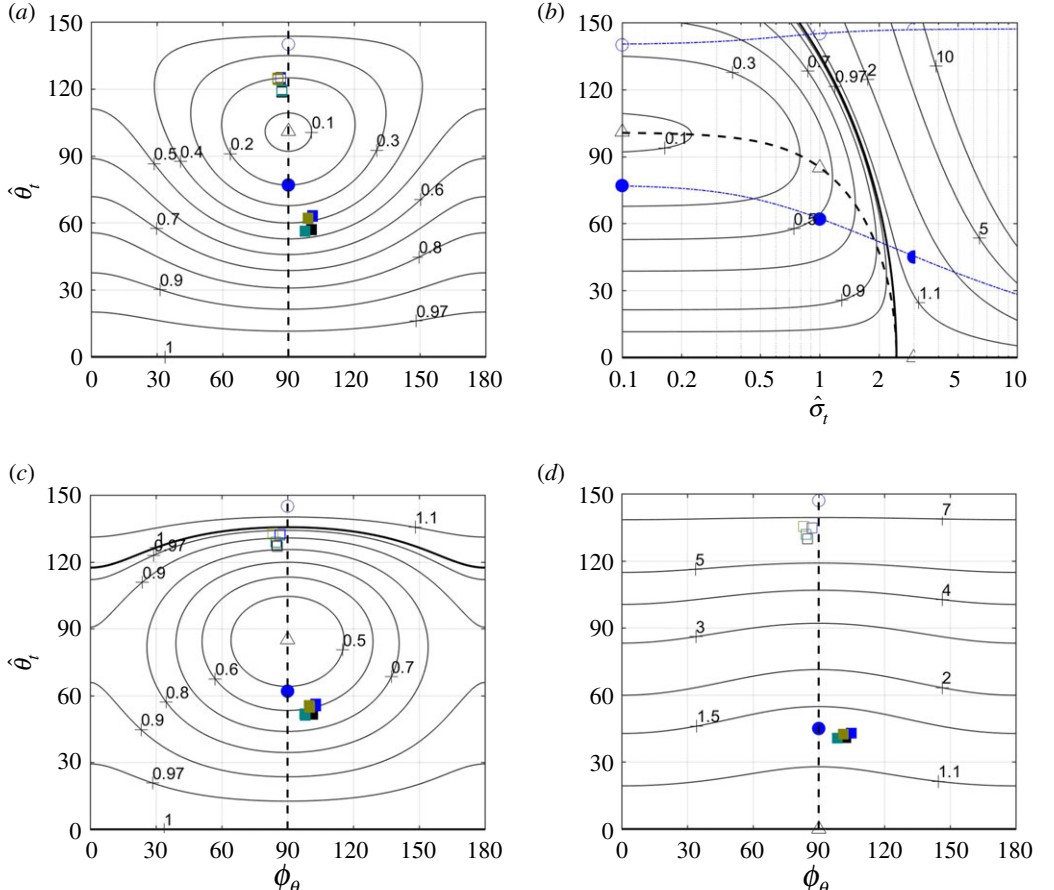

**Figure 2.** Contours of constant $\bar{T}(\phi_\theta,\hat{\theta}_t,0.1)$, $\bar{T}(\phi_\theta,\hat{\theta}_t,1)$ and $\bar{T}(\phi_\theta,\hat{\theta}_t,3)$ over the $\phi_\theta$–$\hat{\theta}_t$ plane are shown on plates (a), (c) and (d), respectively. Contours of constant $\bar{T}(\pi/2,\hat{\theta}_t,\hat{\sigma}_t)$ over the $\hat{\sigma}_t$–$\hat{\theta}_t$ plane are shown on plate (b). The values of $\bar{T}$ along the dashed vertical lines on plates (a), (c) and (d) are found on plate (b) at $\hat{\sigma}_t = 0.1$, $\hat{\sigma}_t = 1$ and $\hat{\sigma}_t = 3$. Thick solid lines on all plates highlight the contours $\bar{T} = 1$. Points and unmarked lines (either dashed or dash-dotted) on all plates show the combinations of parameters where the reduced thrust $\bar{T}$ (triangles, dashed lines), the lift-to-thrust ratio $\bar{L}$ (circles, dash-dotted lines) and the pitching moment-to-thrust ratio $\bar{M}_{z',\text{ref}}$ (squares) are either minimal (empty symbols) or maximal (filled symbols). The values of $\bar{M}_{z',\text{ref}}$ correspond to $x_{\text{ref}} = 0.45$ and $\omega - \kappa = 6$, and the four combinations of shape functions shown in figure 4.

between 101° at $\hat{\sigma}_t \to 0$,[11] and zero at $\hat{\sigma}_t \geq \sqrt{6}$ (figure 2b). The minimum itself ranges from 0.08 at $\hat{\sigma}_t \to 0$ and unity at $\hat{\sigma}_t \geq \sqrt{6}$ (figure 2a,b).

Maxima and minima of $\bar{T}$ should not be confused with maxima and minima of thrust. When swimming at constant speed and depth, thrust equals drag, regardless of the particular value of $\bar{T}$. Smaller $\bar{T}$ merely implies that the swimmer will need larger tail amplitude,

$$\hat{z}_t = \left( \frac{4T}{\pi s_t^2(\omega^2 - \kappa^2)\bar{T}(\phi_\theta,\hat{\theta}_t,\hat{\sigma}_t)} \right)^{1/2}, \tag{5.3}$$

to generate it for the same $\omega$ and $\kappa$ (this equation follows from (5.1)). In fact, since $\bar{T} = 1$ when there is no twist, $\sqrt{1/\bar{T}}$ can be interpreted as the ratio $\hat{z}_t/\hat{z}_{t,\hat{\theta}_t=0}$ of tail amplitudes needed to generate the same thrust when the torsional waves present or not. Having observed $\hat{\theta}_t \approx 50°$—at which $\bar{T}$ can be as low as 0.5 (figure 2a,b), and hence $\hat{z}_t/\hat{z}_{t,\hat{\theta}_t=0}$ can be as high as 1.4—an attempt made in [1] to calculate thrust of a swimming H. platurus based on results of the classical elongated body theory is inconsistent (compare the last two columns in appendix I, table 2).

[11]It is a solution of $2\hat{\theta}_t J_1(2\hat{\theta}_t) = J_2(2\hat{\theta}_t)$.

## 5.2. Lift

Appearing in full in equation (4.11), the expression for lift comprises three terms. The first term, the one involving $\omega - \kappa$, is associated with lift that is (actively) generated by lateral and torsional waves propagating along the tail; it can be compared to the lift generated by a helicopter's rotor in forward flight. The other two are associated with the lift that is (passively) generated by the tail being, on average, at angle with the flow; it can be compared to the lift generated by a wing at angle of attack. The limit $\dot{\hat{z}}_t = \dot{\hat{y}}_t = 0$ leaves only the 'active' part,

$$L = \frac{\pi}{2} s_t^2 J_1(2\hat{\theta}_t)(\omega - \kappa)\hat{z}_t \sin\phi_\theta, \tag{5.4}$$

which is linked to generation of thrust through the product $(\omega - \kappa)\hat{z}_t$. In fact, given $\omega$ and $\kappa$, thrust actually sets $\hat{z}_t$ (equation (5.3)); and hence equation (5.4) can be used to obtain the lift-to-thrust ratio,

$$\frac{L}{T} = \bar{L}(\phi_\theta, \hat{\theta}_t, \hat{\sigma}_t)\left(\frac{\omega - \kappa}{\omega + \kappa}\frac{\pi s_t^2}{T}\right)^{1/2}, \tag{5.5}$$

in which

$$\bar{L}(\phi_\theta, \hat{\theta}_t, \hat{\sigma}_t) = \frac{J_1(2\hat{\theta}_t)\sin\phi_\theta}{\sqrt{\bar{T}(\phi_\theta, \hat{\theta}_t, \hat{\sigma}_t)}} \tag{5.6}$$

manifests the effect of the torsional waves.

The maximum of $\bar{L}$ is practically unity when $\hat{\sigma}_t \to 0$ (figure 3), but vanishes when $\hat{\sigma}_t \to \infty$, when thrust is generated by the torsional waves only. The maximum is invariably associated with $\phi_\theta = \pi/2$ (figure 3a) and $\hat{\theta}_t$ ranging between 77° when $\hat{\sigma}_t \to 0$ and zero when $\hat{\sigma}_t \to \infty$ (figure 3b). With small $\hat{\sigma}_t$ (say, a few tenths), the tail amplitude that will be needed to generate thrust at maximal lift-to-thrust ratio is approximately twice the tail amplitude needed to generate the same thrust with no twist ($\bar{T} \approx 1/4$ along the dash-dotted line with filled circles on figure 2b).

Twisting the tail beyond $\theta_{J_1=0} \approx 110°$ (where $J_1(2\theta) = 0$) makes the lift negative (figure 3a). Its minimum (at $\phi_\theta = \pi/2$ and $\hat{\theta}_t > 140°$) hardly exceeds half of its maximum (at $\phi_\theta = \pi/2$ and $\hat{\theta}_t < 77°$) by the absolute value, and hence negative lift can be generated much more effectively with a smaller twist and $\phi_\theta = -\pi/2$.

## 5.3. Pitching moment

Like the expression (4.11) for lift, the full expression (4.13) for pitching moment comprises three terms, of which the first one (involving $\omega - \kappa$) is associated with moment that is actively generated by lateral and torsional waves propagating along the tail, and the other two are associated with the moment that is passively generated by the tail being, on average, at angle with the flow. This time, however, the limit $\dot{\hat{z}}_t = \dot{\hat{\theta}}_t = \dot{\hat{y}}_t = 0$ leaves two terms in the expression for the pitching moment, that, when re-referred to some $x = x_{\text{ref}}$, take on the form

$$M_{z',\text{ref}} = \frac{\pi s_t^2}{2}\hat{z}_t\left((\omega - \kappa)\sin\phi_\theta\left(J_1(2\hat{\theta}_t)(x_t - x_{\text{ref}}) - X_1(\hat{\theta}_t)\right) - \cos\phi_\theta X_2(\hat{\theta}_t)\right) \tag{5.7}$$

where $X_1(\hat{\theta}_t)$ and $X_2(\hat{\theta}_t)$ shorthand

$$X_1\{\bar{s}, \bar{z}_0, \bar{\theta}_0\}(\hat{\theta}_t) = \int_{x_n}^{x_t} \bar{s}^2(x)\bar{z}_0(x)J_1\left(2\bar{\theta}_0(x)\hat{\theta}_t\right)\,\mathrm{d}x \tag{5.8}$$

and

$$X_2\{\bar{s}, \bar{z}_0, \bar{\theta}_0\}(\hat{\theta}_t) = X_1\{\bar{s}, \dot{\bar{z}}_0, \bar{\theta}_0\}(\hat{\theta}_t), \tag{5.9}$$

whereas $\bar{s}(x) = s(x)s_t^{-1}$, $\bar{z}_0(x) = \hat{z}_0(x)\hat{z}_t^{-1}$ and $\bar{\theta}_0(x) = \hat{\theta}_0(x)\hat{\theta}_t^{-1}$ are the respective shape functions on $(x_n, x_t)$ into $(0, 1)$. Equation (5.7) straightforwardly follows from (3.32) by (4.13) and (4.11). A few examples elucidating the behaviour of $X_1$ and $X_2$ for the shape functions shown in figure 4 (see appendix J for details) can be found in figure 5.

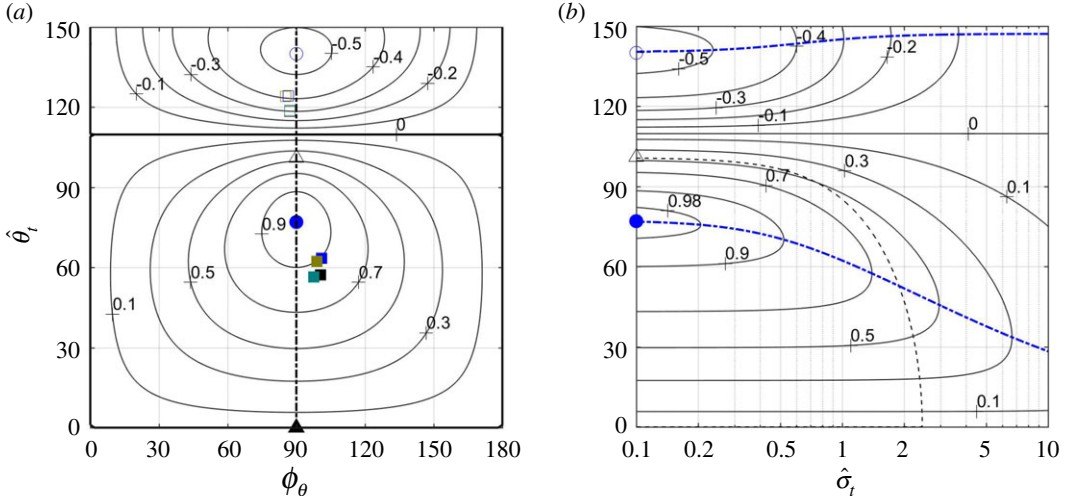

**Figure 3.** Contours of constant $\bar{L}(\phi_\theta, \hat{\theta}_t, 0.1)$ over the $\phi_\theta - \hat{\theta}_t$ plane (a), and of constant $\bar{L}(\pi/2, \hat{\theta}_t, \hat{\sigma}_t)$ over the $\hat{\sigma}_t - \hat{\theta}_t$ plane (b). The values of $\bar{L}$ along the dash-dotted vertical line on plate (a) are those found along the y-axis on plate (b). Thick solid lines on plate (a) highlight the contour $\bar{L} = 0$. Points and unmarked lines (either dashed or dash-dotted) on all plates show the combinations of parameters where the reduced thrust $\bar{T}$ (triangles, dashed lines), the lift-to-thrust ratio $\bar{L}$ (circles, dash-dotted lines) and the pitching moment-to-thrust ratio $\bar{M}_{z',\text{ref}}$ (squares) are either minimal (empty symbols) or maximal (filled symbols). The values of $\bar{M}_{z',\text{ref}}$ correspond to $x_{\text{ref}} = 0.45$ and $\omega - \kappa = 6$, and the four combinations of shape functions shown in figure 4.

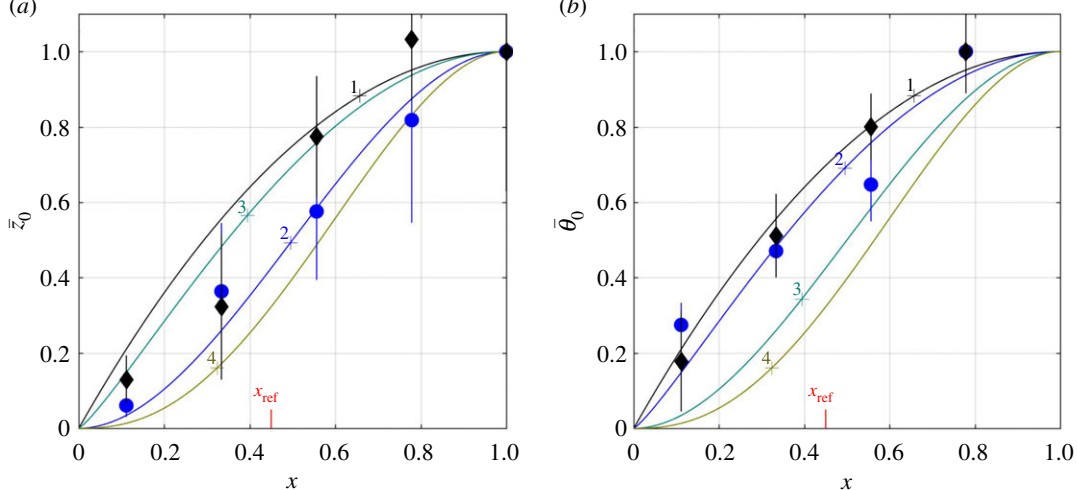

**Figure 4.** The four lines (numbered 1–4) depict four different combinations of $\bar{z}_0$ (a) and $\bar{\theta}_0$ (b) from appendix J. Filled circles and diamonds (with the respective error bars) mark the amplitudes observed with *H. platurus* swimming at 0.3 and 0.6 body lengths per second, respectively [1].

Using (5.3) for $\hat{z}_t$, equation (5.7) furnishes the pitching-moment-to-thrust ratio,

$$\frac{M_{z',\text{ref}}}{T} = \bar{M}_{z',\text{ref}}(\phi_\theta, \hat{\theta}_t, \hat{\sigma}_t, \omega - \kappa)\left(\frac{\omega - \kappa}{\omega + \kappa}\frac{\pi s_t^2}{T}\right)^{1/2}, \tag{5.10}$$

practically in the same form as equation (5.5), only now

$$\bar{M}_{z',\text{ref}}(\phi_\theta, \hat{\theta}_t, \hat{\sigma}_t, \omega - \kappa) = \left((x_t - x_{\text{ref}})J_1(2\hat{\theta}_t) - X_1(\hat{\theta}_t) - \cot\phi_\theta\frac{X_2(\hat{\theta}_t)}{\omega - \kappa}\right)\frac{\sin\phi_\theta}{\sqrt{\bar{T}(\phi_\theta, \hat{\theta}_t, \hat{\sigma}_t)}} \tag{5.11}$$

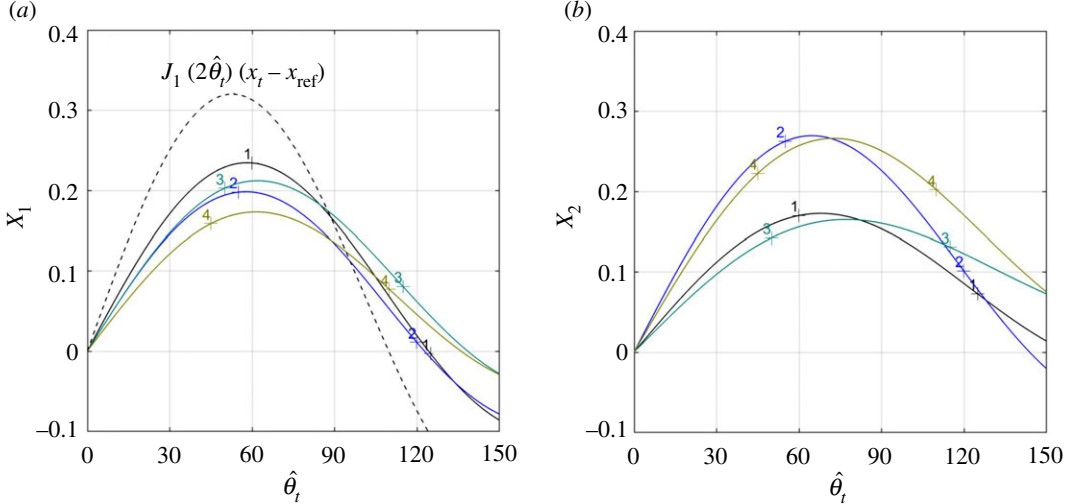

**Figure 5.** $X_1(\hat{\theta}_t)$ and $X_2(\hat{\theta}_t)$ as functions of $\hat{\theta}_t$. $(x_t - x_{\text{ref}})J_1(2\hat{\theta}_t)$ is shown by a broken line on (a) for a reference. Numbers next to each line mark different combination of the shape functions from figure 4, $\bar{s}(x) = \sqrt{x}$ and $x_{\text{ref}} = 0.45$.

replaces $\bar{L}$ as effect of torsion. $\bar{M}_{z',\text{ref}}$ has two extrema,

$$\bar{M}^{\pm}_{z',\text{ref}}(\hat{\sigma}_t, \omega - \kappa) = \pm \left( \frac{\left((x_t - x_{\text{ref}})J_1(2\theta^{\pm}) - X_1(\theta^{\pm})\right)^2}{\bar{T}(\pi/2, \theta^{\pm}, \hat{\sigma}_t)} + \frac{X_2^2(\theta^{\pm})}{(\omega - \kappa)^2 \bar{T}(0, \theta^{\pm}, \hat{\sigma}_t)} \right)^{1/2}, \qquad (5.12)$$

both situated along the line $\phi_\theta = \phi_M(\hat{\theta}_t, \hat{\sigma}_t, \omega - \kappa)$,

$$\phi_M(\hat{\theta}_t, \hat{\sigma}_t, \omega - \kappa) = \arctan\left( (\omega - \kappa) \frac{(x_t - x_{\text{ref}})J_1(2\hat{\theta}_t) - X_1(\hat{\theta}_t)}{-X_2(\hat{\theta}_t)} \frac{\bar{T}(0, \hat{\theta}_t, \hat{\sigma}_t)}{\bar{T}(\pi/2, \hat{\theta}_t, \hat{\sigma}_t)} \right), \qquad (5.13)$$

where $\partial \bar{M}_{z',\text{ref}}/\partial \phi_\theta = 0$. One of the two extrema is a (positive) maximum at $\hat{\theta}_t = \theta^+(\hat{\sigma}_t, \omega - \kappa)$, where both $(x_t - x_{\text{ref}})J_1(2\theta^+) - X_1(\theta^+)$ and $X_2(\theta^+)$ are positive, and hence $\phi_M(\theta^+ \ldots) > \pi/2$; the other is a (negative) minimum at $\hat{\theta}_t = \theta^-(\hat{\sigma}_t, \omega - \kappa)$, where $(x_t - x_{\text{ref}})J_1(2\theta^-) - X_1(\theta^-)$ is negative, and (typically) $\phi_M(\theta^- \ldots) < \pi/2$ (appendix H). Over-twisting the tail causes the anterior and posterior parts of the body to generate lift in the opposite directions, and changes the direction of the pitching moment; unlike the lift, the magnitude of the negative pitching moment exceeds the magnitude of the positive one (figure 6a,c). In those cases, the centre of lift,

$$x_{\text{cL}} = x_{\text{ref}} + \frac{\bar{M}_{z',\text{ref}}}{\bar{L}}, \qquad (5.14)$$

moves posteriad of the caudal end (figure 7).

For all combinations of shape functions tested for this study, $(x_t - x_{\text{ref}})J_1(2\theta^{\pm}) - X_1(\theta^{\pm})$ and $X_2(\theta^{\pm})$ were comparable quantities, and so were $\bar{T}(\pi/2, \theta^{\pm}, \hat{\sigma}_t)$ and $\bar{T}(0, \theta^{\pm}, \hat{\sigma}_t)$. At the same time, $\omega - \kappa$ can be a fairly large quantity—in fact, *H. platurus* swim with $\omega - \kappa \approx 6$ (appendix I, table 2). Consequently,

$$\bar{M}^{\pm}_{z',\text{ref}}(\hat{\sigma}_t, \infty) = \frac{(x_t - x_{\text{ref}})J_1(2\theta^{\pm}) - X_1(\theta^{\pm})}{\sqrt{\bar{T}(\pi/2, \theta^{\pm}, \hat{\sigma}_t)}}, \qquad (5.15)$$

$$\phi_M(\theta^{\pm}(\hat{\sigma}_t, \infty), \hat{\sigma}_t, \infty) = \frac{\pi}{2} \qquad (5.16)$$

and

$$\theta^{\pm}(\hat{\sigma}_t, \infty) = \arg\max_{\hat{\theta}_t} \left( \pm \frac{(x_t - x_{\text{ref}})J_1(2\hat{\theta}_t) - X_1(\hat{\theta}_t)}{\sqrt{\bar{T}(\pi/2, \hat{\theta}_t, \hat{\sigma}_t)}} \right), \qquad (5.17)$$

which formally are limits of the respective quantities when $\omega - \kappa \to \infty$ (see equations (5.12) and (5.13)), can be effectively used as leading-order approximations when $\omega - \kappa$ is finite (figure 6c,d). In fact, when $\omega - \kappa$ exceeds, say, 4, $\phi_M(\theta^{\pm}(\hat{\sigma}_t, \omega - \kappa), \hat{\sigma}_t, \omega - \kappa)$ remains within 15° of $\pi/2$, and neither $\theta^{\pm}(\hat{\sigma}_t, \omega - \kappa)$ nor $\bar{M}^{\pm}_{z',\text{ref}}(\hat{\sigma}_t, \omega - \kappa)$ change appreciably with $\omega - \kappa$.

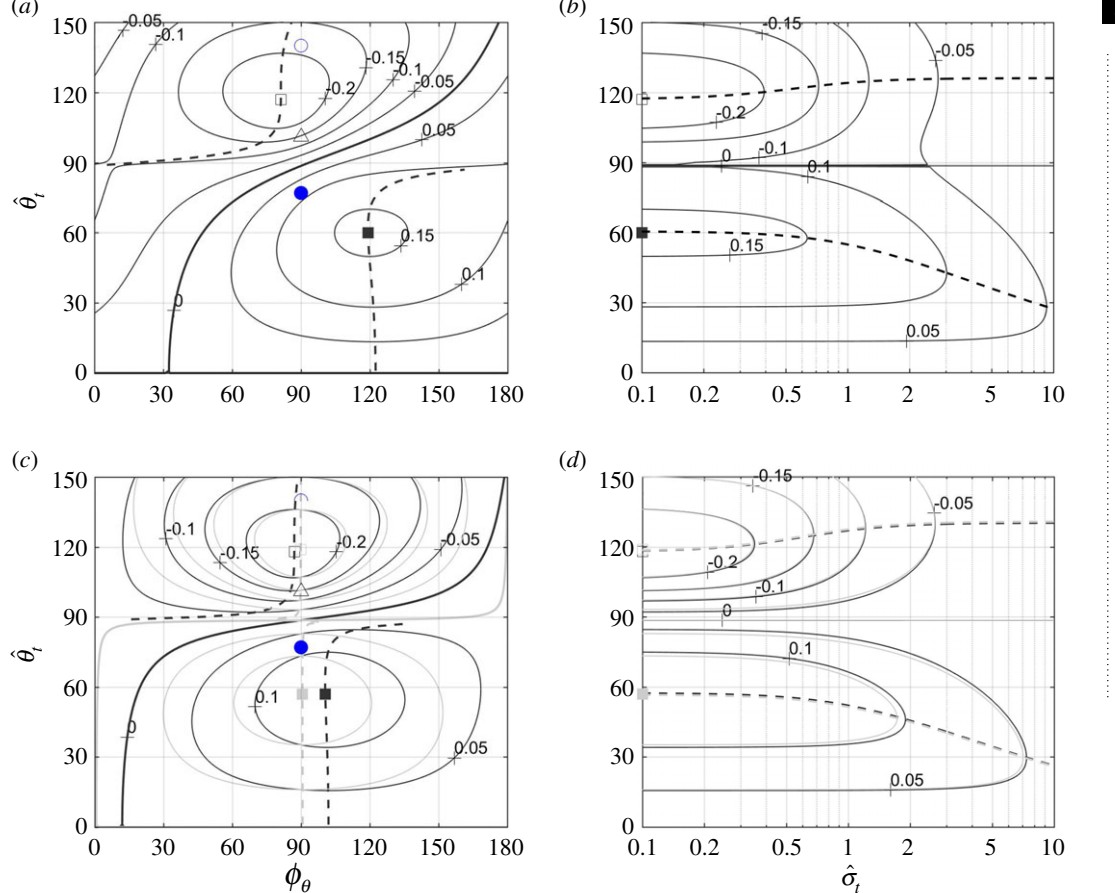

**Figure 6.** Contours of constant $\bar{M}_{z',\mathrm{ref}}(\phi_\theta, \hat{\theta}_t, 0.1, \omega - \kappa)$ over the $\phi_\theta - \hat{\theta}_t$ plane (a,c); contours of constant $\bar{M}_{z',\mathrm{ref}}(\phi_M(\hat{\theta}_t, \hat{\sigma}_t, \omega - \kappa), \hat{\theta}_t, \hat{\sigma}_t, \omega - \kappa)$ over the $\hat{\sigma}_t \, \omega - \hat{\theta}_t$ plane (b,d). On plates (a) and (b), $\omega - \kappa = 2$; on plates (c) and (d), $\omega - \kappa = 6$ (black) and $\omega - \kappa = 100$ (grey). Cases $\omega - \kappa = 6$ and $\omega - \kappa = 100$ are practically indistinguishable on plate (d). $\phi_M(\hat{\theta}_t, 0.1, \omega - \kappa)$ is shown by the dashed lines on plates (a) and (c). The values of $\bar{M}_{z',\mathrm{ref}}$ along those lines are the same as along the y-axes on the respective plates to the right of them. Points mark the combinations of parameters where the reduced thrust $\bar{T}$ (triangles), the lift-to-thrust ratio $\bar{L}$ (circles) and the pitching-moment-to-thrust ratio $\bar{M}_{z',\mathrm{ref}}$ (squares) are either minimal (empty symbols) or maximal (filled symbols). The combination of shape functions underlying this figure is the one that was marked '1' in figure 4, $\bar{s}(x) = \sqrt{x}$, and $x_{\mathrm{ref}} = 0.45$.

Shapes of the modulating functions $\bar{z}_0$ and $\bar{\theta}_0$ have pronounced effect on the pitching moment (figure 8a), but hardly change the arguments $\theta^\pm$ and $\phi_M(\theta^\pm...)$ of its extrema (figure 8b). A combination of late-rising $\bar{z}_0$ and $\bar{\theta}_0$ increases the maximal moment (case 4); a combination of late-raising $\bar{\theta}_0$ and early-rising $\bar{z}_0$ increases (by the absolute value) the minimal one (case 3).

# 6. Balancing a snake

In order to swim at constant depth and speed, thrust should counterbalance drag,[12]

$$T = \pi s_t^2 \bar{D}, \tag{6.1}$$

hydrodynamic lift should counterbalance the excess weight,

$$L = \beta B, \tag{6.2}$$

and the hydrodynamic pitching moment about the centre of mass should counterbalance the hydrostatic

---

[12]In the framework of an ideal fluid approximation, this statement means that the respective tail-beat-averaged component of the force acting on the body through normal and shear stresses on its surface is zero. Thrust and drag has been already associated with normal and shear stresses, respectively.

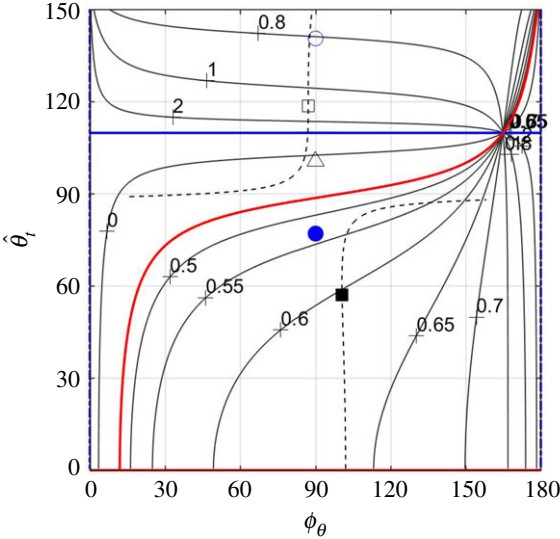

**Figure 7.** Contours of constant centre-of-lift location over the $\phi_\theta$–$\hat{\theta}_t$ plane. Points mark the combinations of parameters where the reduced thrust $\bar{T}$ (triangles), the lift-to-thrust ratio $\bar{L}$ (circles) and the pitching moment-to-thrust ratio $\bar{M}_{z',\text{ref}}$ (squares) are either minimal (empty symbols) or maximal (filled symbols). Solid red line marks the combinations of parameters where $\bar{M}_{z',\text{ref}} = 0$; dot-dashed ('H'-shaped) blue line marks the combinations of parameters where $\bar{L} = 0$; dashed line marks the line $\phi_\theta = \phi_M(\hat{\theta}_t,\hat{\sigma}_t,\omega-\kappa)$ where $\partial\bar{M}_{z',\text{ref}}/\partial\phi_\theta = 0$. The combination of shape functions underlying this figure is the one that was marked '1' in figure 4, $\bar{s}(x) = \sqrt{x}$, $x_{\text{ref}} = 0.45$, $\omega - \kappa = 6$, $\hat{\sigma}_t = 0.1$.

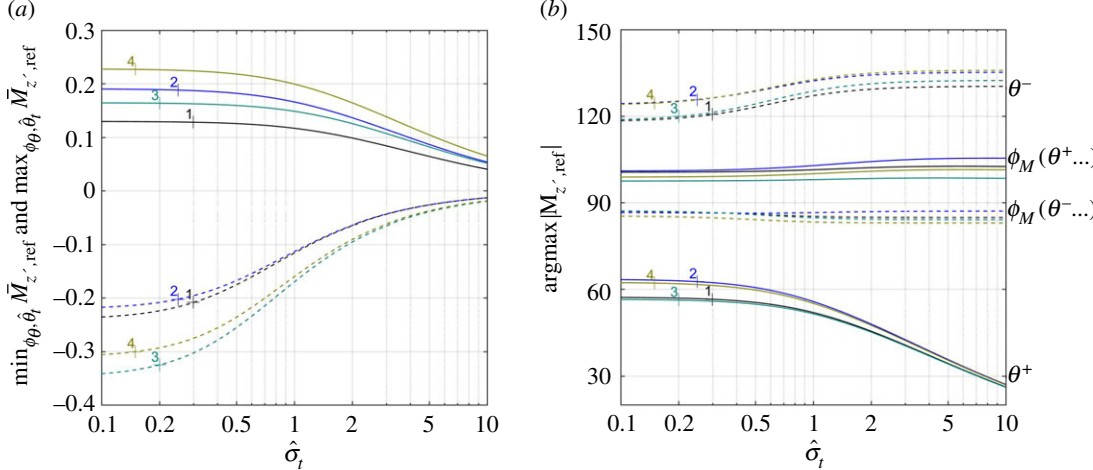

**Figure 8.** Maximal (solid lines) and minimal (dashed lines) pitching moment, $\bar{M}^{\pm}_{z',\text{ref}}(\hat{\sigma}_t,6)$ as functions of $\hat{\sigma}_t$ (a); the respective arg max ($\pm\bar{M}_{z',\text{ref}}$) (b). Numbers next to each line mark different combination of the shape functions (figure 4), $\bar{s}(x) = \sqrt{x}$, $x_{\text{ref}} = 0.45$.

couple

$$M_{z',\text{cm}} = (x_{\text{cm}} - x_{\text{cb}})B. \tag{6.3}$$

In (6.1)–(6.3), $\bar{D}$ is the drag coefficient based on $2\pi s_t^2$ as the reference area; $x_{\text{cb}}$ and $x_{\text{cm}}$ are the respective coordinates of the centres of buoyancy and mass; $B$ is the buoyancy; and $\beta$ is the ratio between the submerged weight and buoyancy.

With $\rho v^2 l^2$ serving as a unit of force (§2.1),

$$B = \frac{\pi s_t^2 k}{\text{Fr}^2}, \tag{6.4}$$

where $k$ is the prismatic coefficient—the ratio between the volume of the body and the minimal cylinder enclosing it (see appendix I)—and Fr is the pertinent Froude number, formally defined

as $\mathrm{Fr} = v/\sqrt{gl}$ ($g$ is the acceleration of gravity). A clear distinction is made here between the actual body shape used for hydrostatic analysis, and its flattened version used for the hydrodynamic one.

With $T$ taken from (6.1), $B$ from (6.4), $L/T$ from (5.5) and $M_{z',\mathrm{cm}}/T$ from (5.10), equilibrium conditions (6.2) and (6.3) can be restated as

$$\bar{L}(\phi_\theta, \hat{\theta}_t, \hat{\sigma}_t)\sqrt{\frac{\omega - \kappa}{\omega + \kappa}\frac{1}{\bar{D}}} = \frac{k\beta}{\mathrm{Fr}^2 \bar{D}} \tag{6.5}$$

and

$$\bar{M}_{z',\mathrm{cm}}(\phi_\theta, \hat{\theta}_t, \hat{\sigma}_t, \omega - \kappa)\sqrt{\frac{\omega - \kappa}{\omega + \kappa}\frac{1}{\bar{D}}} = \frac{k(x_{\mathrm{cm}} - x_{\mathrm{cb}})}{\mathrm{Fr}^2 \bar{D}}. \tag{6.6}$$

Lacking the data, no attempt is made to solve them explicitly. Yet, a necessary condition for their solution to exist is to have the maximal achievable values on their left-hand sides exceed those on the right. Practically, it sets a lower bound on the Froude number. Its rough estimate, based on kinematic data of *H. platurus* (appendix I), yields $1.6 \, |\beta|^{1/2}$ by (6.5), and $3.5 \, |x_{\mathrm{cm}} - x_{\mathrm{cb}}|^{1/2}$ by (6.6). The first one is based on $\bar{L} \approx 1$ (figure 3b); the last one is based on $|\bar{M}_{z',\mathrm{cm}}| \approx 0.2$ (figures 6a and 8a).[13]

A buoyancy–gravity imbalance with $\beta = 0.005$, which is representative of buoyancy loss after an hour at 10 m depth[14] [18,19] or a descent of 1.6 m from the same depth,[15] can be compensated hydrodynamically at Froude numbers in excess of 0.11. This is $0.25 \, \mathrm{m\,s}^{-1}$ for a 0.5 m snake. Sea snakes can swim faster than that. Balancing hydrostatic imbalance hydrodynamically appears as a viable option.

A buoyancy–gravity misalignment with $|x_{\mathrm{cm}} - x_{\mathrm{cb}}| = 0.002$, which is a diminutive 1 mm for a 0.5 m snake, will need a Froude number in excess of 0.15 to be compensated hydrodynamically. This is $0.35 \, \mathrm{m\,s}^{-1}$ for a 0.5 m snake. Sea snakes can swim faster, but it seems unlikely that a realistic hydrostatic couple can be balanced hydrodynamically. Sea snakes do have control over their centre of buoyancy [18].

# 7. Concluding remarks

To make this extension of the elongated (slender) body theory tractable, quite a few simplifying assumptions were made. The central ones were: (i) the body is flat; (ii) it ends at the widest section; (iii) its dorsal and ventral edges both serve as leading edges along their entire length at all times; (iv) the lateral deformations are small; and (v) the Reynolds number is high. The central results are found in equations (3.11)–(3.14), (3.19), (3.26)–(3.34) and (4.10)–(4.14). They were shown coherent in electronic supplementary material, S1 by matching numerical simulations based on the vortex lattice method. Nonetheless, the vortex lattice method cannot serve as a standard to establish their practical applicability limits. To find the limits, the verifying simulations should have been free from any of the assumptions underlying the present results—in particular, free from an *a priori* classification of the swimmers edges into 'leading' and 'trailing' (assumption (iii)). Unsteady RANS simulations could have been effective to this end, but they are complex and deserve a separate study. An encouraging indication of viability of the present results is furnished in appendix I (table 2) by accurately predicting the observed tail amplitude of a swimming *H. platurus*.

Data accessibility. All data underlying this study have been taken from [1,18,19].
Authors' contributions. Both authors took equal share in the analysis; G.I. wrote the manuscript with the input from A.R. Both authors gave final approval for publication.
Competing interests. The authors declare no competing interests.
Funding. This research had no funding.

[13]The reference point for $M_{z',\mathrm{ref}}$ was set at 0.45 body length from the cranial end, roughly where the centre-of-mass is expected to be (appendix I).

[14]To remain neutrally buoyant, a yellow-bellied sea snake needs lungs occupying 6.5% of its body volume [19]. When submerged, it uses oxygen from the lungs and expels $CO_2$ to the water through the skin, losing lungs volume at the rate of approximately 16% per hour at atmospheric pressure [20]. The respective change in buoyancy is approximately 1% per hour. With the same lungs volume at 10 m depth, the rate halves.

[15]Referring to the preceding footnote, a 0.5% change in buoyancy requires 8% change in lungs volume. It can be achieved by changing the outside pressure by the same amount. At 10 m, it requires a change in depth of 1.6 m.

# Appendix A. Normal vectors

A normal to the surface

$$z' = z'_b(t, x, y'), \tag{A 1}$$

can be found with

$$\mathbf{N}'(t, x', y') = \nabla'\left(z' - z'_b(t, x', y')\right) \tag{A 2}$$

(e.g. [20]). When $z'_b$ is given by (2.9), it yields

$$\mathbf{N}'(t, x', y') = \mathbf{e}_{z'} - \mathbf{e}_{y'} \tan\theta(t, x') - \mathbf{e}_{x'}\left(\frac{\partial z_0(t, x')}{\partial x'} + \frac{y' - y_0(t, x')}{\cos^2\theta(t, x')}\frac{\partial\theta(t, x')}{\partial x'} - \tan\theta(t, x')\frac{\partial y_0(t, x')}{\partial x'}\right), \tag{A 3}$$

or, using a different parametrization

$$\mathbf{N}(t, x, y) = \mathbf{e}_{z'} - \mathbf{e}_{y'}\tan\theta(t, x) - \mathbf{e}_{x'}\left(\frac{\partial z_0(t, x)}{\partial x} + \frac{y}{\cos\theta(t, x)}\frac{\partial\theta(t, x)}{\partial x} - \tan\theta(t, x)\frac{\partial y_0(t, x)}{\partial x}\right); \tag{A 4}$$

recall that $\mathbf{e}_x = \mathbf{e}_{x'}$ and $x' = x$ by (2.1) and (2.4), whereas $y'$ and $y$ are related by (2.5) with $z = 0$. When $\partial y_0/\partial x$, $\partial z_0/\partial x$ and $s\partial\theta/\partial x$ are small,

$$|\mathbf{N}(t, x, y)| = |\sec\theta(t, x)| + \ldots \tag{A 5}$$

by (A 4), and in this case, the unit normal, $\mathbf{n} = \pm\mathbf{N}/|\mathbf{N}|$, facing the same side as the z-axis of C, can be recast as a sum

$$\mathbf{n}(t, x, y) = \mathbf{n}_0(t, x) + y\mathbf{n}_1(t, x) + \ldots, \tag{A 6}$$

where

$$\mathbf{n}_0(t, x) = \mathbf{e}_{z'}\cos\theta(t, x) - \mathbf{e}_{y'}\sin\theta(t, x) - \mathbf{e}_{x'}\left(\cos\theta(t, x)\frac{\partial z_0(t, x)}{\partial x} - \sin\theta(t, x)\frac{\partial y_0(t, x)}{\partial x}\right) \tag{A 7}$$

and

$$\mathbf{n}_1(t, x) = -\mathbf{e}_{x'}\frac{\partial\theta(t, x)}{\partial x}; \tag{A 8}$$

note that $\mathbf{e}_{x'} = \mathbf{e}_x$ by (2.1), whereas $\mathbf{e}_{z'}\cos\theta(t, x) - \mathbf{e}_{y'}\sin\theta(t, x) = \mathbf{e}_z$ by (2.2) and (2.3). The ellipsis in (A 5) and (A 6) stands for second-order terms with respect to $\partial y_0/\partial x$, $\partial z_0/\partial x$ and $s\partial\theta/\partial x$.

A unit normal to the dorsal (marked by a plus) and ventral (marked by a minus) edges of the swimmer can be found from the cross product

$$\mathbf{n}_\pm(t, x) = \pm\mathbf{n}(t, x, \pm s(x)) \times \frac{\mathbf{T}_\pm(t, x)}{|\mathbf{T}_\pm(t, x)|}, \tag{A 9}$$

between the unit normal to the surface near the respective edge, and the unit tangent vector to the same edge, $\mathbf{T}_\pm(t, x)/|\mathbf{T}_\pm(t, x)|$;

$$\mathbf{T}_\pm(t, x) = \mathbf{e}_{z'}\frac{\partial\left(z_0(t, x) \pm s(x)\sin\theta(t, x)\right)}{\partial x} + \mathbf{e}_{y'}\frac{\partial\left(y_0(t, x) \pm s(x)\cos\theta(t, x)\right)}{\partial x} + \mathbf{e}_{x'}. \tag{A 10}$$

Written in explicit form, the expression for $\mathbf{n}_\pm(t, x)$ is unwieldy, but when $\partial y_0/\partial x$, $\partial z_0/\partial x$ and $s\partial\theta/\partial x$ are small, it reduces to

$$\mathbf{n}_\pm(t, x) = -\left(\pm\sin\theta(t, x)\frac{\partial z_0(t, x)}{\partial x} \pm \cos\theta(t, x)\frac{\partial y_0(t, x)}{\partial x} + \frac{\mathrm{d}s(x)}{\mathrm{d}x}\right)\mathbf{e}_{x'}$$
$$\pm\cos\theta(t, x)\,\mathbf{e}_{y'} \pm \sin\theta(t, x)\,\mathbf{e}_{z'} + \ldots. \tag{A 11}$$

The ellipsis in (A 11) stands for the same order terms as in (A 5) and (A 6).

# Appendix B. Leading and trailing edges

A point on an edge having coordinates $x' = x$, $y' = y_0(t, x) \pm s(x)\cos\theta(t, x)$ and $z' = z_0(t, x) \pm s(x)\sin\theta(t, x)$ (see equations (2.4)–(2.6)) moves relative to the fluid with velocity

$$\mathbf{v}_\pm(t,x) = -\mathbf{e}_{x'} + \left( \frac{\partial y_0(t,x)}{\partial t} \mp s(x)\sin\theta(t,x)\frac{\partial\theta(t,x)}{\partial t} \right)\mathbf{e}_{y'} + \left( \frac{\partial z_0(t,x)}{\partial t} \pm s(x)\cos\theta(t,x)\frac{\partial\theta(t,x)}{\partial t} \right)\mathbf{e}_{z'}. \quad \text{(B1)}$$

The first term is the velocity of C′ relative to quiescent fluid (i.e. the swimming velocity); the other two are the velocity of the point relative to C′ (it follows by differentiating the coordinates with respect to time). Leading edges of the swimmer are parts of the edges that advance *into* the fluid, i.e. where

$$\mathbf{n}_\pm(t,x) \cdot \mathbf{v}_\pm(t,x) \geq 0; \quad \text{(B2)}$$

trailing edges are the remaining parts. In principle, a part of an edge can be 'leading' during a part of the tail-beat cycle, and 'trailing' during the rest of it. By observation, the flow separates from trailing edges, but not from leading ones.

# Appendix C. Impermeability condition

Defining the surface of the body by a variant $z' - z'_b(t,x',y') = 0$ of (2.7), the impermeability condition on its surface can be formulated as

$$\lim_{z' \to z'_b(t,x',y')} \left( \frac{\partial}{\partial t} + \left( \mathbf{e}_{x'} + \nabla'\phi'(t,x,y',z') \right) \cdot \nabla' \right) \left( z' - z'_b(t,x',y') \right) = 0, \quad \text{(C1)}$$

where the operator in the parentheses is an explicit form of the convective (Lagrangian) derivative [13],[16] whereas $(x',y')$ spans the domain on which the body surface is defined. It is not specified here explicitly, because a different parametrization will eventually be used. Identifying $\nabla'(z' - z'_b(t,x',y'))$ with a normal to the body surface, $\mathbf{N}'(t,x',y')$ (equation (A2)), equation (C1) can be rewritten as

$$\lim_{z' \to z'_b(t,x',y')} \nabla'\phi'(t,x',y',z') \cdot \mathbf{n}'(t,x',y') = \frac{1}{|\mathbf{N}'(t,x',y')|}\left( \frac{\partial}{\partial t} + \frac{\partial}{\partial x'} \right) z'_b(t,x',y'), \quad \text{(C2)}$$

where the sign is to be adjusted to make the respective unit normal $\mathbf{n}' = \pm\mathbf{N}'/|\mathbf{N}'|$ pointing to the same side as the z-axis of C.

The expression on the left of (C2) can be identified as the normal-to-the-surface component of the perturbation velocity. In the leading order with respect to the slenderness parameter, it is $\lim_{z\to +0} \partial\phi(t,x,y,z)/\partial z$, where, having x, y and z related with $x'$, $y'$ and $z'$ by (2.4)–(2.6), $\phi(t, x, y, z) = \phi'(t, x', y', z')$. The expression on the right of (C2) yields

$$\frac{1}{|\mathbf{N}'(t,x',y')|}\left( \frac{\partial}{\partial t} + \frac{\partial}{\partial x'} \right) z'_b(t,x',y') = \cos\theta(t,x)\left( \frac{\mathrm{D}z_0(t,x)}{\mathrm{D}t} - \tan\theta(t,x)\frac{\mathrm{D}y_0(t,x)}{\mathrm{D}t} + \frac{y'-y_0(t,x)}{\cos^2\theta(t,x)}\frac{\mathrm{D}\theta(t,x)}{\mathrm{D}t} \right) \quad \text{(C3)}$$

by (A5), (2.9) and (2.4). Noting that for the problem at hand, $z' \to z'_b(t, x', y')$ implies $z \to 0$, $y' - y_0(t, x) = y\cos\theta(t, x)$ by (2.5). Introducing it in (C3) furnishes the expression that appears on the right-hand side of (2.16).

---

[16]$\mathbf{e}_{x'} + \nabla'\phi'(t,x,y',z')$ is the fluid velocity in C′ by definition of the perturbation potential.

# Appendix D. Pressure jump

Perhaps the easiest way to derive equation (2.31) is by using the inertial reference frame C′. In this frame, the local pressure $p$ can be directly related to the (perturbation) velocity potential by a variant

$$p'(t, x', y', z') = p_\infty + \frac{1}{2} - \frac{\partial \phi'(t, x', y', z')}{\partial t} - \frac{1}{2}\left(\mathbf{e}_{x'} + \nabla'\phi'(t, x', y', z')\right)^2$$

$$= p_\infty - \frac{\partial \phi'(t, x', y', z')}{\partial t} - \frac{\partial \phi'(t, x', y', z')}{\partial x'} - \frac{1}{2}\left(\nabla'\phi'(t, x', y', z')\right)^2 \tag{D1}$$

of Bernoulli's theorem [13]. $p_\infty$ is the pressure far from the moving body; it is reminded that $l$, $v$, $l/v$, $lv$ and $\rho v^2$ are units of length, velocity, time, potential and pressure, respectively. The perturbation potential here can be replaced by

$$\phi'(t, x', y', z') = \phi(t, x, y, z), \tag{D2}$$

where $x = x'$, whereas $y$, $y'$, $z$ and $z'$ are related by the variants of (2.5) and (2.6),

$$y = \sin\theta(t, x)\big(z' - z_0(t, x)\big) + \cos\theta(t, x)\big(y' - y_0(t, x)\big) \tag{D3}$$

and

$$z = \cos\theta(t, x)\big(z' - z_0(t, x)\big) - \sin\theta(t, x)\big(y' - y_0(t, x)\big). \tag{D4}$$

Keeping $y'$ and $z'$ constant when differentiating $\phi$ with respect to $t$ and $x'$, (D 1) yields

$$\begin{aligned}
p(t, x, y, z) = p_\infty &- \frac{D\phi(t, x, y, z)}{Dt} \\
&- \frac{\partial \phi(t, x, y, z)}{\partial y}\left(z\frac{D\theta(t, x)}{Dt} - \sin\theta(t, x)\frac{Dz_0(t, x)}{Dt} - \cos\theta(t, x)\frac{Dy_0(t, x)}{Dt}\right) \\
&- \frac{\partial \phi(t, x, y, z)}{\partial z}\left(-y\frac{D\theta(t, x)}{Dt} - \cos\theta(t, x)\frac{Dz_0(t, x)}{Dt} + \sin\theta(t, x)\frac{Dy_0(t, x)}{Dt}\right) \\
&- \frac{1}{2}\left(\frac{\partial \phi(t, x, y, z)}{\partial y}\right)^2 - \frac{1}{2}\left(\frac{\partial \phi(t, x, y, z)}{\partial z}\right)^2 - \frac{1}{2}\Bigg\{\frac{\partial \phi(t, x, y, z)}{\partial x} \\
&+ \frac{\partial \phi(t, x, y, z)}{\partial y}\left(z\frac{\partial \theta(t, x)}{\partial x} - \sin\theta(t, x)\frac{\partial z_0(t, x)}{\partial x} - \cos\theta(t, x)\frac{\partial y_0(t, x)}{\partial x}\right) \\
&+ \frac{\partial \phi(t, x, y, z)}{\partial z}\left(-y\frac{\partial \theta(t, x)}{\partial x} - \cos\theta(t, x)\frac{\partial z_0(t, x)}{\partial x} + \sin\theta(t, x)\frac{\partial y_0(t, x)}{\partial x}\right)\Bigg\}^2.
\end{aligned} \tag{D5}$$

Because $\phi(t, x, y, z)$ is antisymmetric with respect to $z$ (equation (2.11)), and because

$$\mu(t, x, y) = \lim_{z \to 0}\big(\phi(t, x, y, z) - \phi(t, x, y, -z)\big) \tag{D6}$$

by (2.12), the pressure jump,

$$\Delta p(t, x, y) = \lim_{z \to 0}\big(p(t, x, y, z) - p(t, x, y, -z)\big), \tag{D7}$$

yields (2.31) by (D 5); quadratic terms vanish identically.

# Appendix E. Higher-order pressure moments

Introducing (2.31) in (2.32) one finds

$$\Pi_n(t, x) = \int_{-s(x)}^{s(x)} \frac{D\mu(t, x, y)}{Dt}y^n \, dy - \int_{-s(x)}^{s(x)} \frac{\partial \mu(t, x, y)}{\partial y}\left(\sin\theta(t, x)\frac{Dz_0(t, x)}{Dt} + \cos\theta(t, x)\frac{Dy_0(t, x)}{Dt}\right)y^n \, dy. \tag{E1}$$

Because the potential jump vanishes at $y = \pm s(x)$ by (2.19), the derivative in the first term can be taken outside the integral sign. Subsequent integration by parts allows to recast it as a combination

$$\begin{aligned}
\Pi_n(t, x) = &-\frac{1}{n+1}\frac{D}{Dt}\big(s^{n+2}(x)\mu_{n+1}(t, x)\big) \\
&- s^{n+1}(x)\left(\sin\theta(t, x)\frac{Dz_0(t, x)}{Dt} + \cos\theta(t, x)\frac{Dy_0(t, x)}{Dt}\right)\mu_n(t, x)
\end{aligned} \tag{E2}$$

of potential jump moments (2.20). With (2.27) and (2.28), it splits into

$$
\begin{aligned}
\Pi_n(t,x) = {} & 2\pi \frac{\bar{\mu}_{n+1}}{n+1} \frac{\mathrm{D}}{\mathrm{D}t}\left(s^{n+2}(x)w_0(t,x)\right) \\
& + 2\pi s^{n+2}(x)\bar{\mu}_n\left(\sin\theta(t,x)\frac{\mathrm{D}z_0(t,x)}{\mathrm{D}t} + \cos\theta(t,x)\frac{\mathrm{D}y_0(t,x)}{\mathrm{D}t}\right)w_1(t,x),
\end{aligned}
\tag{E 3}
$$

when $n$ is even, and

$$
\begin{aligned}
\Pi_n(t,x) = {} & 2\pi \frac{\bar{\mu}_{n+1}}{n+1} \frac{\mathrm{D}}{\mathrm{D}t}\left(s^{n+3}(x)w_1(t,x)\right) \\
& + 2\pi s^{n+1}(x)\bar{\mu}_n\left(\sin\theta(t,x)\frac{\mathrm{D}z_0(t,x)}{\mathrm{D}t} + \cos\theta(t,x)\frac{\mathrm{D}y_0(t,x)}{\mathrm{D}t}\right)w_0(t,x),
\end{aligned}
\tag{E 4}
$$

when $n$ is odd.

# Appendix F. Derivation of (3.9)

The terms on the right of (3.6) are can be regrouped as

$$
\begin{aligned}
f_{x'} = {} & -\pi\frac{\mathrm{D}}{\mathrm{D}t}\left(s^2 w_0\left(\cos\theta\frac{\partial z_0}{\partial x} - \sin\theta\frac{\partial y_0}{\partial x}\right)\right) + \pi s^2 w_0\left(\cos\theta\frac{\partial}{\partial x}\frac{\mathrm{D}z_0}{\mathrm{D}t} - \sin\theta\frac{\partial}{\partial x}\frac{\mathrm{D}y_0}{\mathrm{D}t}\right) \\
& - \pi s^2 w_0\frac{\mathrm{D}\theta}{\mathrm{D}t}\left(\sin\theta\frac{\partial z_0}{\partial x} + \cos\theta\frac{\partial y_0}{\partial x}\right) - \pi s^2 w_0\frac{\partial\theta}{\partial x}\left(\sin\theta\frac{\mathrm{D}z_0}{\mathrm{D}t} + \cos\theta\frac{\mathrm{D}y_0}{\mathrm{D}t}\right) \\
& - \frac{\pi}{8}\frac{\partial\theta}{\partial x}\frac{\mathrm{D}}{\mathrm{D}t}\left(s^4 w_1\right) - \frac{\pi}{16}w_1^2\frac{\partial s^4}{\partial x} - \frac{\pi}{2}w_0^2\frac{\partial s^2}{\partial x} - \pi s^2 w_0 w_1\left(\sin\theta\frac{\partial z_0}{\partial x} + \cos\theta\frac{\partial y_0}{\partial x}\right).
\end{aligned}
\tag{F 1}
$$

In this form, the third term on the right cancels out with the last by (2.18), whereas the remaining ones can be further regrouped to obtain

$$
\begin{aligned}
f_{x'} = {} & -\pi\frac{\mathrm{D}}{\mathrm{D}t}\left(s^2 w_0\left(\cos\theta\frac{\partial z_0}{\partial x} - \sin\theta\frac{\partial y_0}{\partial x}\right)\right) + \pi s^2 w_0\frac{\partial\theta}{\partial x}\left(\sin\theta\frac{\mathrm{D}z_0}{\mathrm{D}t} + \cos\theta\frac{\mathrm{D}y_0}{\mathrm{D}t}\right) \\
& - \pi s^2 w_0\frac{\partial\theta}{\partial x}\left(\sin\theta\frac{\mathrm{D}z_0}{\mathrm{D}t} + \cos\theta\frac{\mathrm{D}y_0}{\mathrm{D}t}\right) - \frac{\pi}{2}\frac{\partial}{\partial x}\left(s^2 w_0^2\right) + \frac{\pi}{8}\frac{\mathrm{D}}{\mathrm{D}t}\left(s^4\frac{\mathrm{D}\theta}{\mathrm{D}t}\frac{\partial\theta}{\partial x}\right) - \frac{\pi}{16}\frac{\partial}{\partial x}\left(s^2\frac{\mathrm{D}\theta}{\mathrm{D}t}\right)^2;
\end{aligned}
\tag{F 2}
$$

the fourth term here is the combination of parts from the second and seventh terms in (F 1). The second term now cancels out with the third, yielding (3.9) by (2.18).

# Appendix G. Derivation of (3.23)

Exploiting (2.17) and (2.18), terms on the right-hand side of (3.22) can be regrouped to obtain

$$
\begin{aligned}
\iota = {} & -\pi\frac{\mathrm{D}}{\mathrm{D}t}\left(s^2 w_0\left(\cos\theta\frac{\partial z_0}{\partial t} - \sin\theta\frac{\partial y_0}{\partial t}\right)\right) + \pi s^2 w_0\left(\cos\theta\frac{\mathrm{D}}{\mathrm{D}t}\frac{\partial z_0}{\partial t} - \sin\theta\frac{\mathrm{D}}{\mathrm{D}t}\frac{\partial y_0}{\partial t}\right) \\
& - \pi s^2 w_0\frac{\mathrm{D}\theta}{\mathrm{D}t}\left(\sin\theta\frac{\partial z_0}{\partial t} + \cos\theta\frac{\partial y_0}{\partial t}\right) - \frac{\pi}{2}s^2\frac{\partial w_0^2}{\partial t} + \pi s^2 w_0\frac{\mathrm{D}\theta}{\mathrm{D}t}\left(\sin\theta\frac{\partial z_0}{\partial t} + \cos\theta\frac{\partial y_0}{\partial t}\right) \\
& - \pi s^2 w_0\left(\cos\theta\frac{\partial}{\partial t}\frac{\mathrm{D}z_0}{\mathrm{D}t} - \sin\theta\frac{\partial}{\partial t}\frac{\mathrm{D}y_0}{\mathrm{D}t}\right) + \frac{\pi}{8}\frac{\mathrm{D}}{\mathrm{D}t}\left(s^4\frac{\partial\theta}{\partial t}\frac{\mathrm{D}\theta}{\mathrm{D}t}\right) - \frac{\pi}{16}s^4\frac{\partial}{\partial t}\left(\frac{\mathrm{D}\theta}{\mathrm{D}t}\right)^2.
\end{aligned}
\tag{G 1}
$$

In this form, the second and the third terms cancel out with the fifth and the sixth, yielding (3.23).

# Appendix H. Derivation of (5.12)

In deriving (5.12), two trigonometric identities, $\sin^2\phi_\theta = 1/(1+\cot^2\phi_\theta)$ and $\cos 2\phi_\theta = (\cot^2\phi_\theta - 1)/(\cot^2\phi_\theta + 1)$, prove useful. Using the former, equation (5.2) can be recast as

$$
\bar{T}(\phi_\theta, \hat{\theta}_t, \hat{\sigma}_t) = \frac{\bar{T}(\pi/2, \hat{\theta}_t, \hat{\sigma}_t) + \cot^2\phi_\theta\,\bar{T}(0, \hat{\theta}_t, \hat{\sigma}_t)}{1+\cot^2\phi_\theta}.
\tag{H 1}
$$

Using the latter, together with (H 1), in (5.11), yields

$$\bar{M}_{z',\text{ref}}(\phi_\theta, \hat{\theta}_t, \hat{\sigma}_t, \omega - \kappa) = \frac{J_1(2\hat{\theta}_t)(x_t - x_{\text{ref}}) - X_1(\hat{\theta}_t) - \dfrac{\cot \phi_\theta}{\omega - \kappa} X_2(\hat{\theta}_t)}{\sqrt{\bar{T}(\pi/2, \hat{\theta}_t, \hat{\sigma}_t) + \cot^2 \phi_\theta \bar{T}(0, \hat{\theta}_t, \hat{\sigma}_t)}}. \tag{H2}$$

If $\phi_\theta$ is defined on $(0, \pi)$, there is no ambiguity in sign in (H 2). Equating zero the derivative of $\bar{M}$ with respect to $\cot \phi_\theta$ furnishes (5.13); substituting it back in (H 2) furnishes

$$\bar{M}_{z',\text{ref}}\left(\phi_M(\hat{\theta}_t, \hat{\sigma}_t, \omega - \kappa), \hat{\theta}_t, \hat{\sigma}_t, \omega - \kappa\right) = \text{sgn}\left((x_t - x_{\text{ref}})J_1(2\hat{\theta}_t) - X_1(\hat{\theta}_t)\right)$$

$$\times \left(\frac{\left((x_t - x_{\text{ref}})J_1(2\hat{\theta}_t) - X_1(\hat{\theta}_t)\right)^2}{\bar{T}(\pi/2, \hat{\theta}_t, \hat{\sigma}_t)} + \frac{X_2^2(\hat{\theta}_t)}{(\omega - \kappa)^2 \bar{T}(0, \hat{\theta}_t, \hat{\sigma}_t)}\right)^{1/2}. \tag{H3}$$

Equation (5.12) is a variant of (H 3) with $\hat{\theta}_t = \theta^\pm(\hat{\sigma}_t, \omega - \kappa)$, the point(s) where the derivative of $\bar{M}_{z',\text{ref}}\left(\phi_M(\hat{\theta}_t, \hat{\sigma}_t, \omega - \kappa), \hat{\theta}_t, \hat{\sigma}_t, \omega - \kappa\right)$ with respect to $\hat{\theta}_t$ vanish.

The integrands in (5.8) (that define $X_1(\hat{\theta}_t)$) and (5.9) (that define $X_2(\hat{\theta}_t)$) are positive throughout the integration domain for any $\hat{\theta}_t < \theta_{J_1=0}$ (the angle where $J_1(2\hat{\theta}_t) = 0$, approx. 110°), and hence $X_1(\hat{\theta}_t)$ and $X_2(\hat{\theta}_t)$ change sign only at some $\hat{\theta}_t > \theta_{J_1=0}$. At the same time, the product $(x_t - x_{\text{ref}})J_1(2\hat{\theta}_t)$ changes sign exactly at $\theta_{J_1=0}$, and hence $(x_t - x_{\text{ref}})J_1(2\hat{\theta}_t) - X_1(\hat{\theta}_t)$ changes sign earlier, at some $\hat{\theta}_t < \theta_{J_1=0}$. It implies that $\bar{M}_{z',\text{ref}}\left(\phi_M(\theta^\pm, \hat{\sigma}_t, \omega - \kappa), \theta^\pm, \hat{\sigma}_t, \omega - \kappa\right)$ is a positive maximum if $\theta^\pm$ happens to be smaller than $\theta_{J_1=0}$, where both $(x_t - x_{\text{ref}})J_1(2\theta^\pm) - X_1(\theta^\pm)$ and $X_2(\theta^\pm)$ are positive, and a negative minimum if $\theta^\pm$ happens to be sufficiently large to have the first one negative. Concurrently, the maximum is invariably associated with $\phi_M > \pi/2$, and, in most cases (where $X_2(\theta^\pm)$ is still positive), the minimum is associated with $0 < \phi_M < \pi/2$. The '+' and '−' modifiers with $\theta^\pm$ will be naturally associated with maximum $\bar{M}_{z',\text{ref}}^+$ and minimum $\bar{M}_{z',\text{ref}}^-$ of $\bar{M}_{z',\text{ref}}$, respectively.

# Appendix I. Yellow-bellied sea snake

Graham *et al.* [1] furnishes basic morphological data for seven yellow-bellied sea snakes, 50–70 cm long (the binomial name of this snake has changed twice since this paper was published; it is now *Hydrophis platurus*). An average snake has half-width $s_t = 0.0145$, surface (wet) area $S_w = 0.08$, and prismatic coefficient $k = 0.44$. The same paper also furnishes kinematic data (recapitulated in the first five columns of table 2) for a 0.51 m snake at two swimming speeds, 0.15 and 0.32 m s$^{-1}$, but does not provide any additional information on the particular snake for which the data was collected. The missing data were supplemented by assuming average values. The phase lag, $\phi_\theta$, was guessed to be larger than 90° (say, 120°) based on the general comment made in this reference that '…the keel flared outward at maximal displacement…'. Drag coefficient of the snake at the two speeds was estimated with $\bar{D} = S_w C_f(\text{Re})/2\pi s_t^2$, where $C_f(\text{Re}) \approx 0.455(\log_{10}\text{Re})^{-2.58}$ is an empirical approximation for the effective friction coefficient [11] and Re is the pertinent body-length-based Reynolds number; recall that the drag was associated with the viscous constituent only. It was tacitly assumed that textured skin of the snake renders the boundary layer turbulent. $\bar{D}$ and $C_f$ values in table 2 confirm the values estimated in table 4 of [1]. $\hat{z}_{t,\hat{\theta}_t=0}$ and $\hat{z}_t$ were estimated with (5.3) assuming that thrust equals drag and $\hat{\sigma}_t = s_t/\hat{z}_t = 0$ (the estimate does not change when changing $\hat{\sigma}_t$ to 0.1). It practically recovers the value reported in [1]—compare the 4th and 13th columns in table 2.

**Table 2.** Swimming parameters of a 0.51 m yellow-bellied sea snake. The bottom line of the table specifies the source of the preceding two lines.

| $v$ m s$^{-1}$ | $\Omega$ | $\kappa$ | $\hat{z}_t$ | $\hat{\sigma}_t$ | $\hat{\theta}_t$ | $\phi_\theta$ | Fr | Re $10^3$ | $C_f 10^{-3}$ | $\bar{D}$ | $\hat{z}_{t,\hat{\theta}_t=0}$ | $\hat{z}_t$ |
|---|---|---|---|---|---|---|---|---|---|---|---|---|
| 0.15 | 17.8 | 12.5 | 0.129 | 0.11 | 51° | 120° | 0.067 | 77 | 7.6 | 0.458 | 0.107 | 0.138 |
| 0.32 | 17.1 | 11.6 | 0.121 | 0.12 | 45° | 120° | 0.14 | 164 | 6.4 | 0.387 | 0.099 | 0.121 |
| | | | ref. [1] | | | a guess | | | | | (112) | (112) |

**Table 3.** Coefficients $c_1$, $c_2$ and $c_3$ that generate $\bar{z}_0$, $\bar{\theta}_0$ and $\bar{s}$ with (J 1).

| function | case 1 | | | case 2 | | | case 3 | | | case 4 | | |
|---|---|---|---|---|---|---|---|---|---|---|---|---|
| | $c_1$ | $c_2$ | $c_3$ | $c_1$ | $c_2$ | $c_3$ | $c_1$ | $c_2$ | $c_3$ | $c_1$ | $c_2$ | $c_3$ |
| $\bar{z}_0$ | 1 | 2 | 0 | 2 | 3 | 0 | 1.2 | 2.2 | 0 | 2.5 | 3.5 | 0 |
| $\bar{\theta}_0$ | 1 | 2 | 0 | 1.2 | 2.2 | 0 | 2 | 3 | 0 | 2.5 | 3.5 | 0 |
| $\bar{s}$ | 1/2 | 0 | 1/2 | 1/2 | 0 | 1/2 | 1/2 | 0 | 1/2 | 1/2 | 0 | 1/2 |

# Appendix J. Shape functions

To allow visualization of the pitching moment and the functions associated with it, one needs the shape functions $\bar{z}_0(x) = \hat{z}_0(x)\hat{z}_t^{-1}$, $\bar{\theta}_0(x) = \hat{\theta}_0(x)\hat{\theta}_t^{-1}$ and $\bar{s}(x) = s(x)/s_t$. All of them were generated by fitting $c_1 \geq 0$, $c_2 \geq 0$ and $c_3$ in

$$f(x; c_1, c_2, c_3) = \frac{c_2 - c_3}{c_2 - c_1}x^{c_1} - \frac{c_1 - c_3}{c_2 - c_1}x^{c_2}. \tag{J1}$$

For any viable combination of $c_1$, $c_2$ and $c_3$, this function implicitly satisfies $f(0; c_1, c_2, c_3) = 0$, $f(1; c_1, c_2, c_3) = 1$ and $\lim_{x \to 1}\partial f(x; c_1, c_2, c_3)/\partial x = c_3$. The particular parameters of $\bar{z}_0$ and $\bar{\theta}_0$ (table 3) were chosen so as to have the observations of Graham *et al.* [1] bracketed between the limiting cases (figure 4), and make the respective derivatives $\dot{\bar{z}}_0$ and $\dot{\bar{\theta}}_0$ vanish at the tail section. In all cases, the body outline was assumed parabolic, with $\bar{s}(x) = \sqrt{x}$.

Lungs of *H. platurus* occupy the entire body length up to the flattened tail section [19]. The length of that section is approximately 11% body length (table 1 in [1]). Based on these observations, the reference point for calculation of the pitching moment $x_{\text{ref}}$ was chosen at 0.45 (body lengths from the cranial end).

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
