## [Reviewer comments · Royal Society Open Science]

Review History

RSOS-200754.R0 (Original submission)

Review form: Reviewer 1

Is the manuscript scientifically sound in its present form?

Yes

Are the interpretations and conclusions justified by the results?

Yes

Is the language acceptable?

Yes

Do you have any ethical concerns with this paper?

No

Have you any concerns about statistical analyses in this paper?

No

Recommendation?

Accept with minor revision (please list in comments)

Comments to the Author(s)

The paper is a valuable contribution to the literature. It is not surprising perhaps that torsional waves and angle of attack can generate lift, but the detailed calculations are challenging and I commend the authors on performing them. It would be interesting in a future work to consider what body deformations are optimal for some measure of swimming performance.

Some specific comments:

"Can it be that the torsional wave comes to balance the swimming snake in the vertical plane?"--it would be good to say explicitly what it would balance against--weight/gravity/negative buoyancy.

Some typos, awkward language. E.g.

"slender body theory relies of three ...", "makes solution for the forces bulky"-bulky--cumbersome?

Eqn. 14-- please include a reference.

Please provide references to the biological literature to support the assertion "Deformations of a swimming snake or a moray eel can be adequately represented by a combination of lateral and torsional waves". How much twist is possible for these animals given their anatomical constraints?

Review form: Reviewer 2

Is the manuscript scientifically sound in its present form?

Yes

Are the interpretations and conclusions justified by the results?

Yes

Is the language acceptable?

Yes

Do you have any ethical concerns with this paper?

No

Have you any concerns about statistical analyses in this paper?

No

Recommendation?

Major revision is needed (please make suggestions in comments)

Comments to the Author(s)

The manuscript considers an extension of the slender body theory to three-dimensional deformations of an elongated, thin (ribbon-like) structure. The authors solve the potential flow around the structure, as well as the hydrodynamic forces and moments acting on it, in the limit of small lateral deformations and superimposed torsional waves. The goal is to analyze the effect of the combination of lateral and torsional waves on the lift and thrust forces acting on the structure, in order to shed light on the propulsion modes of sea snakes.

The topic is interesting and the approach is technically sound (although I did not re-derive the equations myself). I think the work is valuable. However, I have several suggestions to improve the manuscript itself:

1- Since the motivation stems from sea snakes, I think the authors should start with a schematic depiction of sea snakes and their undulation modes, as well as clearly-labeled schematic of the model. I understand that Fig. 1 is intended to serve the purpose of explaining the model but the figure is obscure.

2. The model assumptions, although properly explained, do not seem intuitive. Why choose an increasing profile that ends with two shed trails at the dorsal and ventral sides, rather than a geometrically decreasing profile from head to tail ending with a single shed trail (or two)? Is the physics that is being captured here dependent on the difference between a dorsal and a ventral trail?

3. The paper is extremely heavy on mathematical notation and symbols. Although the authors try to give intuition and proper explanation when appropriate, the manuscript could be made easier to read if the authors include a table with proper explanation of the nomenclature and symbols, including a table of non-dimensional parameters. For example, the authors set the non-dimensional velocity $U = 1$ and refer to thrust T as power, when they mean TU (with $U=1$). I suggest keeping TU for easier readability.

4. Most importantly, I think that the physical insight regarding the effect of the coupling between lateral deformations and twist on lift and thrust is lost at the end of the manuscript, hidden in somewhat cumbersome and small plots, with unintuitive labels on the axes. I suggest that the authors label the axes by words first followed by the symbol, for example, "time, t " instead of simply " t ". This would make the results more readable and accessible even for readers who do not wish to follow the mathematical derivations.

In sum, I think that this work is promising but the manuscript could benefit from a thorough edit.

Decision letter (RSOS-200754.R0)

Dear Professor Iosilevskii,

The editors assigned to your paper ("Hydrodynamics of a twisting slender swimmer") have now received comments from reviewers. We would like you to revise your paper in accordance with the referee and Associate Editor suggestions which can be found below (not including confidential reports to the Editor). Please note this decision does not guarantee eventual acceptance.

Please submit a copy of your revised paper before 28-Jun-2020. Please note that the revision deadline will expire at 00.00am on this date. If we do not hear from you within this time then it will be assumed that the paper has been withdrawn. In exceptional circumstances, extensions may be possible if agreed with the Editorial Office in advance. We do not allow multiple rounds of revision so we urge you to make every effort to fully address all of the comments at this stage. If deemed necessary by the Editors, your manuscript will be sent back to one or more of the original reviewers for assessment. If the original reviewers are not available, we may invite new reviewers.

To revise your manuscript, log into <http://mc.manuscriptcentral.com/rsos> and enter your Author Centre, where you will find your manuscript title listed under "Manuscripts with

Decisions." Under "Actions," click on "Create a Revision." Your manuscript number has been appended to denote a revision. Revise your manuscript and upload a new version through your Author Centre.

- Data accessibility

If you wish to submit your supporting data or code to Dryad (<http://datadryad.org/>), or modify your current submission to dryad, please use the following link:
<http://datadryad.org/submit?journalID=RSOS&manu=RSOS-200754>

- Competing interests

- Authors' contributions

- Acknowledgements

- Funding statement

on behalf of Dr Kenta Ishimoto (Associate Editor) and Mark Chaplain (Subject Editor)
openscience@royalsociety.org

Comments to Author:

Reviewers' Comments to Author:

Reviewer: 1

Comments to the Author(s)

The paper is a valuable contribution to the literature. It is not surprising perhaps that torsional waves and angle of attack can generate lift, but the detailed calculations are challenging and I commend the authors on performing them. It would be interesting in a future work to consider what body deformations are optimal for some measure of swimming performance.

Some specific comments:

"Can it be that the torsional wave comes to balance the swimming snake in the vertical plane?"--it would be good to say explicitly what it would balance against--weight/gravity/negative buoyancy.

Some typos, awkward language. E.g.

"slender body theory relies of three ...", "makes solution for the forces bulky"-bulky--cumbersome?

Eqn. 14-- please include a reference.

Please provide references to the biological literature to support the assertion "Deformations of a swimming snake or a moray eel can be adequately represented by a combination of lateral and torsional waves". How much twist is possible for these animals given their anatomical constraints?

Reviewer: 2

Comments to the Author(s)

The manuscript considers an extension of the slender body theory to three-dimensional deformations of an elongated, thin (ribbon-like) structure. The authors solve the potential flow around the structure, as well as the hydrodynamic forces and moments acting on it, in the limit of small lateral deformations and superimposed torsional waves. The goal is to analyze the effect of the combination of lateral and torsional waves on the lift and thrust forces acting on the structure, in order to shed light on the propulsion modes of sea snakes.

The topic is interesting and the approach is technically sound (although I did not re-derive the equations myself). I think the work is valuable. However, I have several suggestions to improve the manuscript itself:

- 1- Since the motivation stems from sea snakes, I think the authors should start with a schematic depiction of sea snakes and their undulation modes, as well as clearly-labeled schematic of the model. I understand that Fig. 1 is intended to serve the purpose of explaining the model but the figure is obscure.
2. The model assumptions, although properly explained, do not seem intuitive. Why choose an increasing profile that ends with two shed trails at the dorsal and ventral sides, rather than a geometrically decreasing profile from head to tail ending with a single shed trail (or two)? Is the physics that is being captured here dependent on the difference between a dorsal and a ventral trail?
3. The paper is extremely heavy on mathematical notation and symbols. Although the authors try to give intuition and proper explanation when appropriate, the manuscript could be made easier to read if the authors include a table with proper explanation of the nomenclature and symbols, including a table of non-dimensional parameters. For example, the authors set the non-dimensional velocity $U = 1$ and refer to thrust T as power, when they mean TU (with $U=1$). I suggest keeping TU for easier readability.
4. Most importantly, I think that the physical insight regarding the effect of the coupling between lateral deformations and twist on lift and thrust is lost at the end of the manuscript, hidden in somewhat cumbersome and small plots, with unintuitive labels on the axes. I suggest that the authors label the axes by words first followed by the symbol, for example, "time, t " instead of simply " t ". This would make the results more readable and accessible even for readers who do not wish to follow the mathematical derivations.

In sum, I think that this work is promising but the manuscript could benefit from a thorough edit.

Author's Response to Decision Letter for (RSOS-200754.R0)

See Appendix A.

Decision letter (RSOS-200754.R1)

Dear Professor Iosilevskii,

It is a pleasure to accept your manuscript entitled "Hydrodynamics of a twisting slender swimmer" in its current form for publication in Royal Society Open Science.

You can expect to receive a proof of your article in the near future. Please contact the editorial office (openscience_proofs@royalsociety.org) and the production office

(openscience@royalsociety.org) to let us know if you are likely to be away from e-mail contact -- if you are going to be away, please nominate a co-author (if available) to manage the proofing process, and ensure they are copied into your email to the journal.

Best regards,

on behalf of Dr Kenta Ishimoto (Associate Editor) and Mark Chaplain (Subject Editor)
openscience@royalsociety.org

Appendix A

Reviewer: 1

Comments to the Author(s)

The paper is a valuable contribution to the literature. It is not surprising perhaps that torsional waves and angle of attack can generate lift, but the detailed calculations are challenging and I commend the authors on performing them. It would be interesting in a future work to consider what body deformations are optimal for some measure of swimming performance.

Thank you

Some specific comments:

"Can it be that the torsional wave comes to balance the swimming snake in the vertical plane?"--it would be good to say explicitly what it would balance against--weight/gravity/negative buoyancy.

The first paragraph in the introduction was changed

Some typos, awkward language. E.g. "slender body theory relies of three ...", "makes solution for the forces bulky"-bulky--cumbersome?

We hope that the language was improved throughout

Eqn. 14-- please include a reference.

The entire section was rewritten, and, we hope, properly referenced.

Please provide references to the biological literature to support the assertion "Deformations of a swimming snake or a moray eel can be adequately represented by a combination of lateral and torsional waves". How much twist is possible for these animals given their anatomical constraints?

We think that we have cited all relevant literature, and this question will have to remain with no answer. There are many youtube videos that support this claim, e.g.

The adaptations of sea snakes - The Wonder of Animals: Episode 11 Preview - BBC Four

https://www.youtube.com/watch?v=8E_I4AgsIok

but they cannot be used as formal references. Based on this video, the angle can exceed 90 deg.

Reviewer: 2

Comments to the Author(s)

The manuscript considers an extension of the slender body theory to three-dimensional deformations of an elongated, thin (ribbon-like) structure. The authors solve the potential flow around the structure, as well as the hydrodynamic forces and moments acting on it, in the limit of small lateral deformations and superimposed torsional waves. The goal is to analyze the effect of the combination of lateral and torsional waves on the lift and thrust forces acting on the structure, in order to shed light on the propulsion modes of sea snakes.

The topic is interesting and the approach is technically sound (although I did not re-derive the equations myself). I think the work is valuable.

Thank you

However, I have several suggestions to improve the manuscript itself:

1- Since the motivation stems from sea snakes, I think the authors should start with a schematic depiction of sea snakes and their undulation modes, as well as clearly-labeled schematic of the model. I understand that Fig. 1 is intended to serve the purpose of explaining the model but the figure is obscure.

The figure was redrawn. We hope that it is clearer now.

2. The model assumptions, although properly explained, do not seem intuitive. Why choose an increasing profile that ends with two shed trails at the dorsal and ventral sides, rather than a geometrically decreasing profile from head to tail ending with a single shed trail (or two)? Is the physics that is being captured here dependent on the difference between a dorsal and a ventral trail?

Increasing profile ends with a single wake sheet comprising both vertical and horizontal vortices (assuming that the snake swims straight and level). In the slender body theory, the wake behind the body has no influence on the flow field around the body. This what makes the greatest simplification. We have shown in Section 3.5 that the wake indeed carries the difference between the energy spent and the energy made good. In any case we slightly revised sections 2.2 and 2.3 that introduce the model.

3. The paper is extremely heavy on mathematical notation and symbols. Although the authors try to give intuition and proper explanation when appropriate, the manuscript could be made easier to read if the authors include a table with proper explanation of the nomenclature and symbols, including a table of non-dimensional parameters. For example, the authors set the non-dimensional velocity $U = 1$ and refer to thrust T as power, when they mean TU (with $U=1$). I suggest keeping TU for easier readability.

Nomenclature section was added and so were a few clarifying comments in Section 3.5, where power made good was used.

4. Most importantly, I think that the physical insight regarding the effect of the coupling between lateral deformations and twist on lift and thrust is lost at the end of the manuscript, hidden in somewhat cumbersome and small plots, with unintuitive labels on the axes. I suggest that the authors label the axes by words first followed by the symbol, for example, "time, t " instead of simply " t ". This would make the results more readable and accessible even for readers who do not wish to follow the mathematical derivations.

All the figures were redrawn and Section 5 was rewritten. We hope it is clearer now.

In sum, I think that this work is promising but the manuscript could benefit from a thorough edit.

We sincerely hope that the present version is better, and we thank the referee for the constructive comments.